# Boldness, activity, and aggression: Insights from a large-scale study in Baltic salmon (*Salmo salar* L)

**Johanna Axling[1,2], Laura E. Vossen[3], Erik Peterson[4], Svante Winberg[1,2,3]\***

**1** Behavioural Neuroendocrinology, Department of Medical Cell Biology, Uppsala University, Uppsala, Sweden, **2** Behavioural Neuroendocrinology, Department of Neuroscience, Uppsala University, Uppsala, Sweden, **3** Division of Anatomy and Physiology, Department of Anatomy, Physiology and Biochemistry, Swedish University of Agricultural Sciences, Uppsala, Sweden, **4** Department of Aquatic Resources, Swedish University of Agriculture, Uppsala, Sweden

\* Svante.Winberg@neuro.uu.se

**Data Availability Statement:** Data has been uploaded to https://figshare.com/; DOI 10.6084/m9.figshare.23611272.

## Abstract

Atlantic salmon (*Salmo salar*) display high levels of agonistic behavior in aquaculture farms, resulting in fin damage and chronic stress. Aggression affects fish growth and performance negatively, and presents a serious welfare problem. Indeed, it would be beneficial to identify, separate or exclude overly aggressive individuals. Research on behavioral syndromes suggests that aggressive behavior may correlate with other behavioral traits, such as boldness and locomotory activity. We aimed to develop a high-throughput method to quantify and predict aggressive behavior of individual parr in hatchery-reared Baltic salmon (*Salmo salar* L.). We screened approximately 2000 parr in open field (OF) and mirror image stimulation (MIS) tests. We extracted seven variables from video tracking software for each minute of the tests; distance moved and duration moving (activity), the duration in and number of entries to the center of the arena (boldness), the distance moved in and duration spent in the area adjacent to the mirror during the MIS test (aggressiveness) and head direction (lateralization). To investigate the relationship between activity, boldness and aggression we first correlated the first six variables to one another. Second, we assigned individuals to high, medium, low or zero aggression groups based on the MIS test and quantified activity and boldness in each group. Third, we analyzed whether the fish viewed the mirror with the left or right eye. Our results show that medium and low aggressive fish were the most active, while highly aggressive fish showed average activity. Aggressive groups did not differ in boldness. Activity and boldness were positively correlated. Finally, we detected a preference for fish to view the mirror with the left eye. We conclude that aggressiveness cannot be predicted from the results of the OF test alone but that the MIS test can be used for large-scale individual aggression profiling of juvenile salmon.

**Funding:** Formas (ANIHWA 2014-01842 to SW) website: https://formas.se/en/start-page.html The Swedish Research Council (VR 2017-03779 to SW) website: https://www.vr.se/english.html FACIAS foundation (to SW and JA) website:https://stiftelsemedel.se/stiftelsen-facias/ The funders had no role in study design, data collection and analysis, decision to publish, or preparation of the manuscript.

**Competing interests:** The authors have declared that no competing interests exist.

## Introduction

The health and welfare of fish in aquaculture systems is of growing concern to consumers [1, 2]. In aquatic species such as fish, health and welfare are tightly linked to water quality, stocking density and disease control [3, 4]. In addition, aggression between individuals is a major cause of stress and fin damage [5], which often leads to bacterial or fungal infections commonly seen in hatcheries [6]. Considering that teleost fish possess nociceptors and show withdrawal from potentially painful stimuli [7], it is likely that fish perceive fin damage as painful, which raises ethical concerns. Furthermore, secondary effects of fin damage on fish health may comprise production efficiency, quality and quantity [8].

Atlantic salmon (*Salmo salar* L) is an important aquaculture species [9] which shows high levels of aggression in hatcheries. In Sweden and Norway, Baltic populations of Atlantic salmon are also bred in hatcheries to compensate for loss of access to their natural spawning areas due to hydroelectric power plants [10]. During the freshwater phase juvenile salmon are territorial and the level of agonistic behavior is high [6, 11–13]. These agonistic responses can include indirect interaction such as displays, color changes, threatening postures as well as overt aggressive acts, such as chasing and biting [9, 14, 15]. There is evidence that the aggressive display of salmon is lateralized. Fish may view their opponent with a particular eye, although both a right eye and left eye bias have been reported [16]. Displays, both lateral and frontal [17, 18] are aggressive signals that challenge and reinforce dominance without direct physical interactions. During an initial encounter with a conspecific, the fish may switch back and forth between direct attack (overt aggression) and display (passive aggression) [19] before a stable dominance hierarchy is established. Subordinate fish tend to show a stress-induced behavioral inhibition [20] and if possible will try to escape [21, 22].

It has becoming increasingly clear that salmonids and other teleosts display intraspecific divergence in behavioral and physiological responses to challenge [23]. Behavioral responses is often described as personalities, which include boldness, exploration, activity, aggression, and sociability [24]. Physiological responses are often called stress coping and includes physiological traits, such as sympathetic reactivity and post-stress plasma cortisol [25]. Animals displaying a proactive coping style are characterized by being bold, aggressive, showing active avoidance, high sympathetic reactivity but only modest stress-induced elevation of plasma glucocorticoids whereas reactive animals showing the opposite profile being shy, non-aggressive and displaying high post-stress plasma glucocorticoids [23].

To reduce the detrimental effects of aggression in aquaculture, it would be beneficial to identify and possibly separate or exclude overly aggressive individuals, termed "personality profiling" [24]. In fish research, behavioral test of aggression exposing the individual to a mirror have gained in popularity, since such tests reduce the need for repeated testing against multiple opponents [9]. The mirror image stimulation (MIS) test has been used to quantify aggression in salmonids [18, 25–29] as well as other fish species [21, 30]. Even though it is still a matter of debate whether the MIS test accurately reflects dyadic agonistic behavior, salmon presented with a mirror readily react with an aggressive display that resembles the display used in dyadic fights [31, 32]. Holtby et al. [18] tested the response of juvenile coho salmon (*Oncorhynchus kisutch*) in the MIS test and subsequently paired up each animal with an opponent of similar size in a stream tank. The results from the MIS test were a significant predictor of the outcome of agonistic interactions among individuals [18].

While it is labor intensive to assess the aggressiveness of individual fish at an industrial scale, aggression may be related to other behavioral traits that are easier to quantify. Research on animal personalities in fish suggests that aggressive behaviors often correlate with behavioral traits from other contexts, such as boldness and locomotory activity that can be measured

in the widely used open field (OF) test [33, 34]. In a now classical paper, Huntingford et al. [35] reported that in non-reproductive three-spined stickleback (*Gasterosteus aculeatus*) males, boldness was positively correlated with aggression during the breeding season, a relationship which has been repeatedly confirmed for this species previously [36–38]. Additional studies also included locomotor activity as a part of the bold and aggressive behavioral profile [39] where there was a positive relationship between activity, aggression and boldness in juvenile and adult three-spined sticklebacks. However, the relationship between aggression and boldness can differ between populations due to differences in predation pressure [40, 41].

In the current study, we aimed to develop a high-throughput method to quantify and predict the behavioral profile of individual parr of a hatchery-reared stock of Baltic salmon. First, we analyzed the relationship between locomotory activity, boldness, and aggression in approximately 2000 parr with an OF and MIS test. Using a combination of automated video tracking and manual scoring, we selected six behavioral variables and quantified these for each minute of the tests. Distance moved and duration moving were chosen as proxies for activity, the duration in and the number of entries to the center of the arena as proxies for boldness and the distance moved in and duration of time spent in the area closest to the mirror were proxy variables for aggressive behavior. Subsequently, we analyzed these variables in two ways: i) we classified animals into four aggressiveness groups, ranging from high to zero aggression and quantified differences in activity and boldness between aggression groups; ii) we correlated our six tracking variables with one another. Finally, we investigated whether aggressive behavior was lateralized [16], i.e. whether the parr showed a left or right eye bias when viewing the mirror.

Behavioral tests at group level, e.g. response to hypoxia, boldness, have been described for several teleosts. However, when testing fish in groups behavioral responses may be confound by agonistic interactions. Moreover, in salmon there is limited information on how responses to hypoxia and/or boldness determined in groups relates to aggressive behavior of individual fish.

## Material & methods

### Animal breeding and maintenance

Experimental protocols and animal handling methods used in this study were approved by the Uppsala Regional Animal Ethics Board (Dnr C55/13), following the guidelines of the Swedish Legislation on Animal Experimentation (Animal Welfare Act SFS1998:56) and the European Union Directive on the Protection of Animals Used for Scientific Purposes (Directive 2010/63/EU). Adult Baltic salmon (100 females and 100 males) were collected from the river Dalälven in a fish trap right outside of the Älvkarleby fishery research station (60˚ 34' 7.3128" N, 17˚ 26' 7.0944" E) during the spawning season, from July to November 2015.

Milt and roe were collected from the adults in November 2015 and the eggs were kept in horizontal flow-through metal hatching trays in separate perforated compartments (20 × 14 × 17 widths × length × height) with water from the river Dalälven. Baltic salmon suffer from the M74 syndrome resulting in abnormally high yolk-sac fry mortality. The incidence of M74 varies greatly between years [42]. All fry of family groups displaying M74 symptoms were excluded from this study.

In May 2016, 20 to 40 days after hatching and as soon as the fry started to feed, 20 fry from each family group were mixed and moved to two standard hatchery tanks (width × height 106 x 106 cm, water depth 30 cm) made of opaque grey plastic. The photoperiod in the hatchery was automatically adapted to conditions at latitude 60˚ N. The fish density in the tank was approximately 2000 individuals per tank. All tanks were supplied with flow-through, naturally

tempered water from the river Dalälven. Commercial food pellets (Aller Aqua Futura (0,5–1 cm, Christiansfeld, Denmark) were given from automatic feed dispensers (Fry Feeder 907, AquaCultur, Nienburg, Germany) based on water temperature, average weight from all the fish in the tank and fish density (approximately 0.5% of their body mass). On 15–17 August 2016, when the fish were 4 months old, they were anaesthetized with 5% benzocaine (25–50 mg/L; Sigma-Aldrich, Sweden) [43] and passive integrated transponder (PIT) tags (12.5 mm ISO FDX-B, Biomark, USA) were surgically implanted into the ventral cavity. After PIT tagging groups were mixed and were evenly divided over four hatchery tanks (~750 parr per tank).

## Behavioral testing

Behavioral tests were conducted from 30 August until 28 October 2016 in a separate room at the Älvkarleby station. Fish were fasted 24 hours prior to testing which is a standard procedure to avoid problems with regurgitated food. The test arena was a modified white opaque plastic container (model TROFAST, IKEA, Sweden; dimensions 30 × 42 × 23 cm length × width × height) fitted with a mirror on one of the short sides and filled with water to a water depth of 10 cm (Fig 1). Two removable hatches were placed parallel to the mirror (Fig 1), allowing the experimenter to record both the OF and MIS tests within the same arena (adapted from Adriaenssens & Johnsson, [21]). Hatch 1 was placed 29 cm from the mirror and divided the start box from the rest of the arena. Hatch 2 was placed 8 cm from the mirror and ensured that the fish could not see the mirror while the open field test was recorded. Test arenas were filled with water from the river Daläven, and water temperature was recorded for every day and the external temperature was ranging from 16.5 (August) to 5.6°C (end of October). When the fish were tested, they weighed 10.4 ± 0.09 g and had standardized fork length 9.28 cm ± 0.03 cm (mean ± SEM, n = 1987).

Upon behavioral testing, an individual fish was netted out of the hatchery tank, the PIT tag code was recorded with an ID plate reader (Biomark, USA) and the fish was released into the start box of the test arena. We video recorded 16 fish (1 fish per arena) simultaneously using four surveillance cameras (IB8369, Vivotek, New Taipei city, Taiwan), each camera filming four arenas. In total 137 trials were performed. Cameras were mounted on the ceiling and were controlled from outside of the behavioral recording room. The recording was started directly after the last fish was released into the start box. Five minutes after start of the

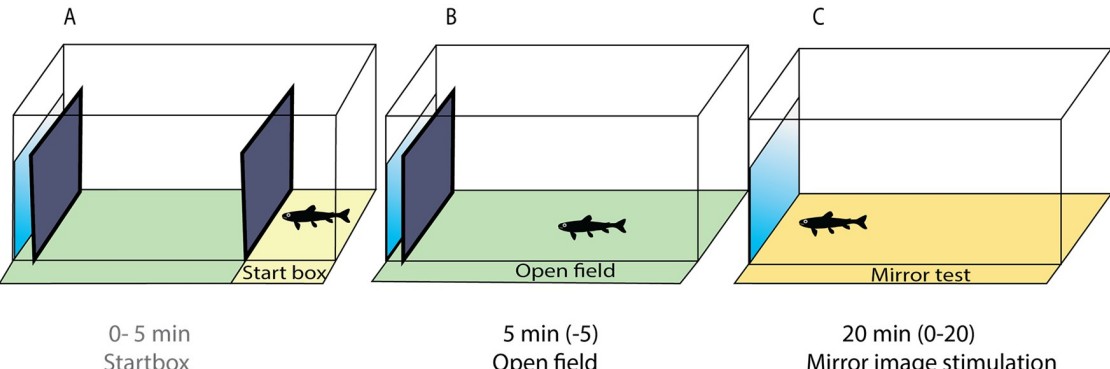

**Fig 1. A-C**: Schematic representation of the test arena showing the timing of the tests. A) After 5 minutes in the start box hatch 1 was removed and (B) the open field (OF) test started, which lasted 5 minutes. C) Subsequently hatch 2 was removed, the mirror became visible, and the fish was video recorded for a final 20 minutes in the mirror image stimulation (MIS) test.

recording, hatch 1 was removed from all arenas and the OF test started, which lasted for 5 minutes. Ten minutes after the recording started, hatch 2 was removed and the MIS test started, after which the fish was left undisturbed for a final 20 minutes. The hatches were pulled out upwards from a hole in the top of the hatch using a long stick with a metal hook, which ensured minimal disturbance by the observer. Between trials, the arenas were cleaned with 70% ethanol and rinsed with water. After behavioral testing, each fish was netted out of the testing arena and was anesthetized with benzocaine solution (5% concentration, 25-50mg/L) [43]. The adipose fin was collected for potential future genomic analyses and the fish were weighed and fork length was recorded. After this the fish was moved to a new hatchery tank where all tested fish from the original hatchery tank were collected. In total 1987 salmon parr underwent behavioral testing.

## Behavioral analysis

All videos were analyzed using automated tracking software EthoVision XT14 (Noldus Technology, Wageningen, The Netherlands) with nose-tail base detection. Two zones of interest were virtually drawn in the arena: a center zone and a mirror zone (Fig 2). We extracted the following variables from the tracking software for each minute: distance moved (in cm) in the whole arena and in the mirror zone, the duration in the center and mirror zone (in sec), the number of entries into the center and mirror zone and the head direction, both while the fish was anywhere in the arena, and specifically while it was in the mirror zone.

Following initial analysis of tracking variables, we manually scored 105 randomly selected fish for the duration spent "striking" against the mirror during minute 5 (05:00–06:00), using BORIS observation software [44]. Striking was defined as a fast movement, resulting from a strong tail beat, towards the mirror at an angle perpendicular to the mirror. The choice for this time interval was based on visual inspection of the variables obtained from Ethovision.

## Statistical analysis

The statistical analyses were performed in SAS statistical software version 9.4 (SAS Institute INC, city, Country) and R statistical computing software version 4.1.1 [45] with additional packages "lmer" [46], "emmeans" [47], "ggplot2" [48], "car" [49] and "robcor" [50].

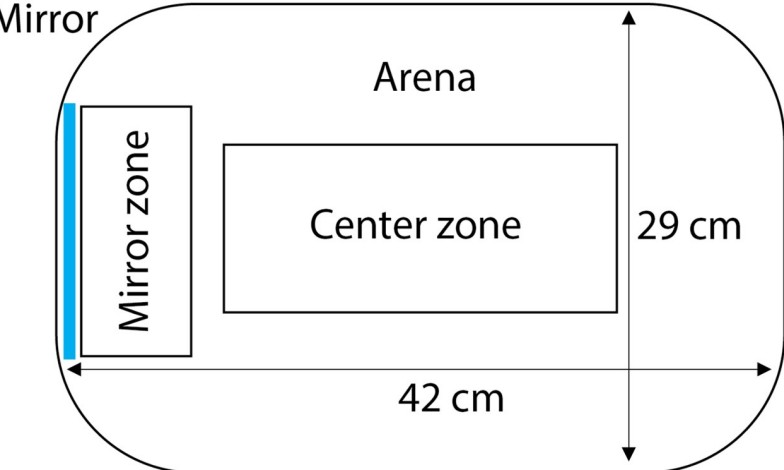

**Fig 2. Schematic representation of the virtual division of the arena into zones used for the video tracking analysis.** The arena was divided into three zones: whole arena, center zone and mirror zone. The image is not drawn on scale.

Initial inspection of the videos revealed that only a proportion of fish actively attacked the mirror, therefore we first aimed to classify individuals into discrete aggressiveness levels. Our most reliable measure of aggressiveness was the manually scored variable "striking", the number of seconds per minute that the individual spent moving perpendicular towards the mirror at a fast speed [9, 14, 51]. This variable was conveniently used to classify the ~5% of fish for which manual scoring was available, into 4 discrete levels: highly aggressive (HA), "striking" for 50–60 sec/min; medium aggressive (MA) 30–50 sec/min; low aggressive (LA) 0.001–30 sec/min; zero aggressive (ZA) 0 sec/min. In order to classify the remaining fish for which manual scoring was not available, we made use of the strong relationship between "striking" and two video tracking variables, the duration of time spent in the mirror zone and the distance moved in the mirror zone (Fig 3). Since the relationship was not linear, we used the 'PROC DISCRIM' function in SAS statistical software, where the manually recorded observations were considered the 'DATA' and the rest of the observations comprising the 'TESTDATA', i.e. the observations to classify. This resulted in a set of probabilities for each individual of belonging to each of the four aggressive groups. We assigned the individual to the group that was associated with the highest probability. By this approach we were able to assign 1679 out of 1987 fish (84%) to an aggression group. However, the rest of the fish (16%) had missing values for one or both of the variables used in the classification (i.e. "duration in mirror zone" and "distance moved in mirror zone") and could not be assigned to an aggression group.

We continued our analyses in R. We first investigated the effect of arena (categorial variable with 16 levels), water temperature (covariate, range 5.7 to 16.4 ˚C), day of testing (covariate, range 1 to 60) and session (categorical variable with 4 levels) on the behavioral variables using univariate linear mixed-effects models (LMMs) with a random intercept effect of fish ID. This revealed an effect of arena and water temperature on activity variables distance moved and duration moving but no effect of session (model results not shown). The effect of day of testing was strongly related to water temperature, with a clear stepwise reduction in behavioral activity variables for temperatures above 15˚C [52]. We fitted six mixed-effects models to the data to investigate whether aggression group affected each of the six behavioral variables measured per minute. Response variables distance moved in the arena, the proportion of time moving (logit transformed), and distance moved in the mirror zone were analyzed with LMMs containing a fixed effect of Aggression group, Minute and their interaction, and random (intercept) effects of fish ID, Arena and Temperature. Since variables did not show linear patterns over the 25 minutes of the behavioral tests, variable Minute was recoded as a categorical factor with 25 levels. Considering the stepwise effect of water temperature on locomotory activity variables, Temperature was included as a random factor with 3 levels (<10˚C, 10–15˚C or >15˚C). Similar LMMs were constructed for response variables proportion of time spent in the center zone (logit transformed), and proportion of time in the mirror zone (logit transformed), although here we did not include a random effect of temperature since there was no clear effect of temperature on the proportion of time spent in the center zone (r2 = 0.0035) or mirror zone (r2 = 0.0017). In both cases we used pooled within-test-minute correlations. Despite those values were significant (p<0.001), we regarded the level of significance as inflated due to multiple comparison, and in addition, the effect size were regarded as too low to be relevant. The number of entries into the center zone was analyzed with a negative binomial generalized linear mixed-effects model (Negative Binomial GLMM, function 'glmer.nb') with the same explanatory variables. Post-hoc pair-wise comparisons were calculated using the package 'emmeans'. For all models, we evaluated assumptions (normality, homoscedasticity, outliers) using residual plots. For the negative binomial GLMM we also assessed overdispersion.

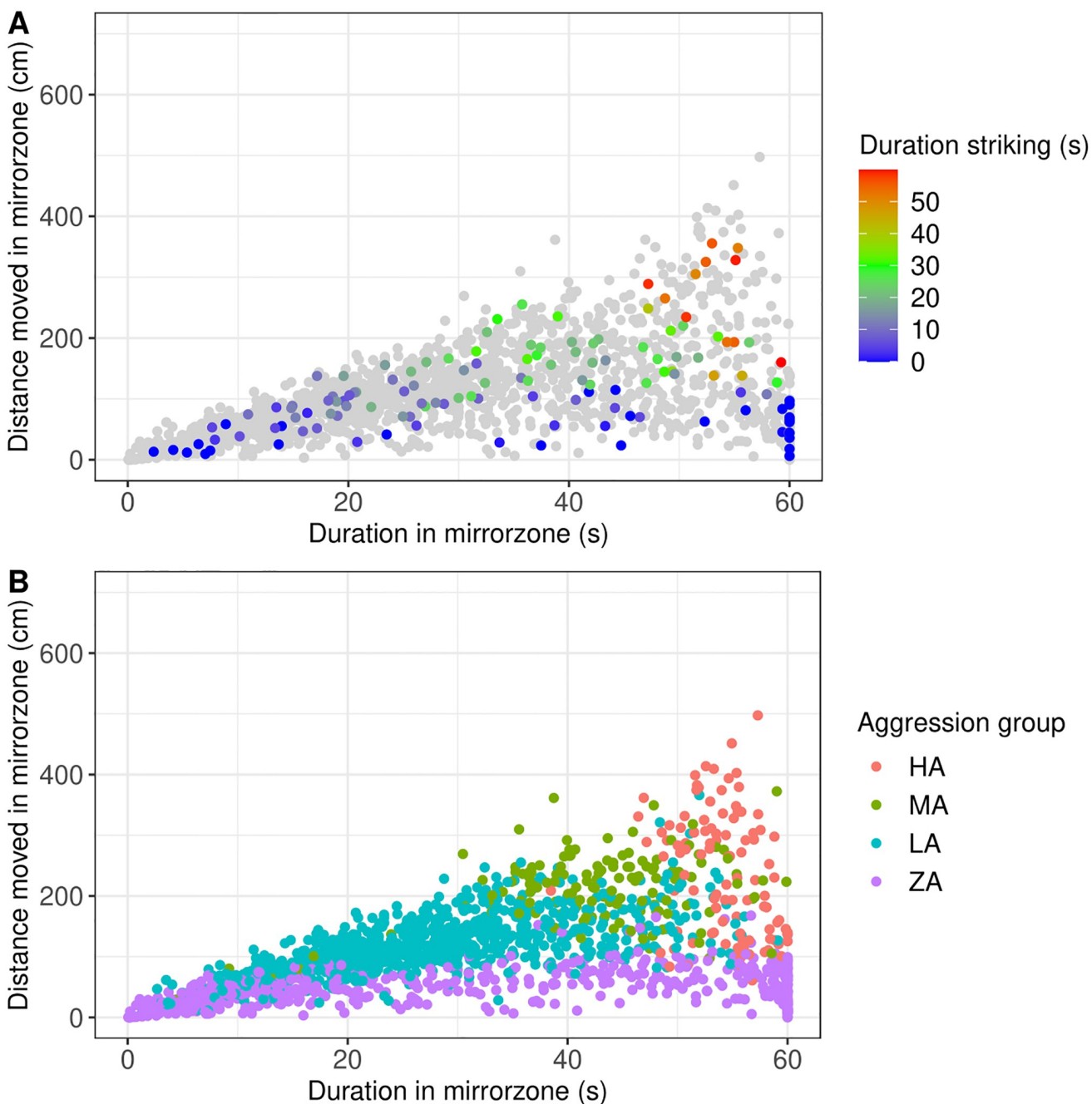

**Fig 3. Relationship between manually scored variables and variables extracted from video tracking software at minute 5 of the MIS test.** Duration in the mirror zone against velocity in the mirror zone for each individual parr. In A) The continuous color scale indicates the number of seconds spent displaying "striking" behavior. The colored dots represent the subset of fish manually scored and the grey dots represents all fish from the study. In B) the discrete color scale indicates the aggression group to which each individual was assigned using discriminant analysis.

We correlated the two activity variables (distance moved in the arena, the proportion of time moving at minute) and two boldness variables (proportion of time spent in the center zone, the number of entries into the center zone) from the OF test (minute -1), with the aggression variables from the MIS test (duration in mirror zone and distance moved in the

mirror zone; measured at minute 5:00). We calculated pairwise Pearson's correlation coefficients (using the 'cor.test' function), as well as robust correlation coefficients using the 'robcor' function from the 'robcor' package.

We recoded the variable head direction while in mirror zone (in degrees) into a categorical factor (8 'octants' of 45 degrees). This made it possible to combine the data from all arenas, regardless of whether the mirror was placed on the left or right short side. We constructed a generalized linear model with binomial error distribution (Binomial GLM) of the proportion of fish in each octant at minute 5, with fixed effects of Octant, Aggression group and Mirror side and all interactions. An equivalent GLM was constructed for the proportion of fish in each octant while the fish was in the mirror zone.

## Results

### Time course of behavior

"Distance moved" increased over time in both the OF test and MIS test (LMM contrast; minute -5 vs. -1, z = 30.165, p = <0.001; minute 1 vs. 19, z = -23.430, p<0.001; Fig 4A). The activity variable "duration moving" in arena showed a similar pattern over time (Fig 4B).

"Duration in center zone" declined over time to reach a low level at minute -1 (LMM contrast minute -5 vs. -1, z = -24.623, p = <0.001) and was at a constant lower level during MIS test. "Frequency in the center zone" also decreased over the OF test (LMM contrast minute -5 vs. -1, z = 16.348, p = <0.001) and continued to decrease during the MIS test until minute 3 (LMM contrast minute 0 vs. 3, z = 7.063 p = <0.001), but increased again during the remaining 15 minutes of the MIS test (LMM contrast minute 3 vs. 10, z = -7.629, p = <0.001; Fig 4D).

During the MIS test, "duration in mirror zone" showed a steep increase from minute 0 to 3 (LMM contrast minute 0 vs. 3, z = -40.879, p = <0.001), which subsided around minute 10 (LMM contrast minute 3 vs. 10, z = 14.728, p<0.001) after which it remained stable for the rest of the MIS test (LMM contrast minute 10 vs. 19, z = 3.453, p = 0.1052; Fig 4E). A similar pattern was observed for the variable "distance moved in mirror zone" with a peak in the beginning of the MIS test (main effect of Minute: LMM, $F_{24,34696}$ = 136.361, p<0.0001; Fig 4F).

The aggression group classification was based on variables duration and distance moved in the mirror zone in minute 5 (see Methods section 2.4). Our mixed models confirmed highly significant differences between the groups consistent across all minutes of both tests for "duration in mirror zone" (LMM, $F_{19,31794}$ = 210.968, p<0.001) and "distance moved in mirror zone" (LMM, $F_{19,29287}$ = 159.298, p<0.001). For minute 4 to 10 of the OF test, group HA spent the longest time in the mirror zone, followed by MA (LMM contrast HA vs MA, min 5, z = 4.764, p<0.001), LA (LMM contrast, HA vs LA: z = 15.760, p<0.001) and lastly ZA (LMM contrast, HA vs. ZA: z = 15.624, p<0.001; Fig 4E). In the MIS test, groups differed in the height and timing of the peak in "duration spent in the mirror zone". Group HA reached a maximum value at minute 6, (LMM contrast HA vs MA, z = 5.418, p<0.001), HA vs LA: z = 17.173, p<0.001, HA vs ZA; z = 14.458, p<0.001), MA showed a lower and blunter peak at minute 3, 4 and 5, LA peaked at minute 4, while ZA reached a maximum at minute 3 and did not show a subsequent decline. (Fig 4E). HA and MA moved greatest "distances in the mirror zone", followed by LA (LMM contrast HA vs LA: z = 14.282, p<0.001; MA vs LA: z = 15.425, p<0.001) and ZA traveled shortest "distances in the mirror zone" (LMM contrast HA vs ZA: z = 22.842, p<0.001; LA vs ZA: z = 17.183, p<0.001 and MA vs ZA: z = 24.807, p<0.001 Fig 4F).

### Differences in activity and boldness between aggression groups

The aggression groups showed highly significant and consistent differences in "distance moved" throughout both the OF and MIS tests (main effect of Group, LMM, $F_{24,40163}$ =

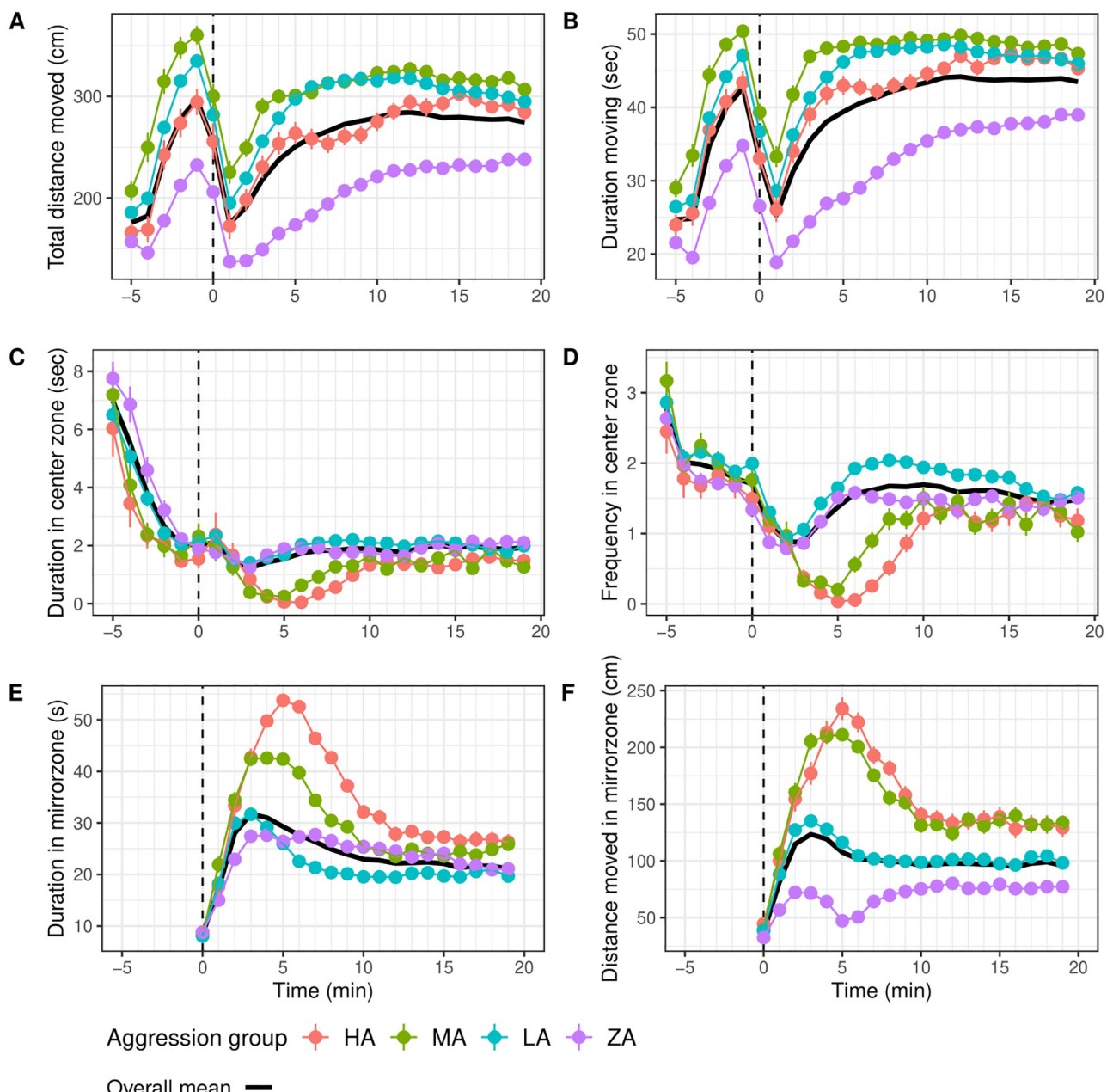

**Fig 4. Video tracking variables in the MIS test for the four aggression groups, as well as the overall mean.** (A) Distance moved in arena (**activity**) (cm). (B) Duration moving in arena (**activity**) (s). (C) Frequency of visits to in center zone (**boldness**). (D) Duration in center zone (**boldness**)(s). (E) Duration in mirror zone (**aggression**)(s). (F) Distance moved in mirror zone (**aggression**)(cm). In each graph, the black line represents the overall mean. The x-axis represents time (in minutes), e.g. -5 indicates minute -05:00 to -04:00. From minute -5 to 0 the arena served as an open field arena, while the dashed vertical line at minute 0 indicates the minute in which hatch 2 was opened and the mirror became visible.

153.305, p<0.001; Table 1). Groups MA and LA moved largest distances (LMM contrast, z = 2.294, p = 0.131), HA moved intermediate distances (LMM contrast, HA vs. LA: z = -3.022, p<0.01; HA vs MA: z = -4.044, p<0.001) and the ZA group was least active (LMM contrast, MA vs ZA: z = 12.430, p<0.001; HA vs ZA: z = 6.346, p<0.001, Fig 4A). The HA group showed a transient decline in "distance moved" between minute 5 and 12. For activity variable

**Table 1. Descriptive statistics of the behavioral variables per aggression group and timepoint.** Minute -1 was the last minute of the OF test, and minute 5 and 10 minutes were 5 and 10 minutes into the MIS test. Values represent means ± SEM. Capital letters in superscript (H, M, L and/or Z) indicate (a) significant difference(s) from other aggression groups, i.e. high (H), medium (M), low (L) or zero (Z) aggression groups, according to post-hoc pair-wise comparisons on mixed-effects models. Note that the values here represent the untransformed values, while some variables were transformed in statistical models, see Methods section.

| | Test minute | Highly aggression | Medium aggression | Low aggression | Zero aggression | Overall |
|---|---|---|---|---|---|---|
| Distance moved (cm) | -5 | 166±9.58 | 207±9.74[Z] | 186±4.18[Z] | 157±4,41[M, L] | 176±115 |
| | -1 | 294±12.74[M,L,Z] | 360±9.23[H,Z] | 335±4.43[H,Z] | 232±6.10[H,M,L] | 297±144* |
| | 5 | 264±11.02[Z] | 301±6.58[Z] | 297±3.52[Z] | 174±5.10[H,M,L] | 251±125* |
| | 10 | 275±9.10[Z] | 323±6.76[Z] | 315±3.08[Z] | 221±4.98[H,M,L] | 279±111* |
| | 19 | 284±8.46[Z] | 307±6.55[Z] | 295±3.04[Z] | 238±4.21[H,M,L] | 274±97.1 |
| Duration moving (s) | -5 | 24±1.44 | 29±1.25[Z] | 26.4±0.52[Z] | 21.5±0.57[M,L] | 24.7±14.8 |
| | -1 | 43.4±1.57[M,L,Z] | 50.4±0.73[H,Z] | 47.1±0.47[H,Z] | 34.8±0,78[H,M,L] | 42.6±16.9* |
| | 5 | 43±1.30[Z] | 48.3±0.63[Z] | 46.2±0.34[Z] | 27.6±0.74[H,M,L] | 39.4±16.2* |
| | 10 | 44.6±1.05[Z] | 49.1±0.55[Z] | 48.2±0.29[Z] | 35.4±0.70[H,M,L] | 43.4±13.8* |
| | 19 | 45.3±0.90[Z] | 47.3±0.61[Z] | 46±0.34[Z] | 39±0.60[H,M,L] | 43.5±12.1 |
| Frequency in center | -5 | 2.45±0.31 | 3.17±0.27 | 2.86±0.12 | 2.63±0.16 | 2.78±3.59 |
| | -1 | 1.68±0.18 | 1.74±0.14 | 1.88±0.07 | 1.68±0.08 | 1.78±2* |
| | 5 | 0.0367±0.03[L,Z] | 0.204±0.05[L,Z] | 1.65±0.08[M] | 2.30±0.09[H,M] | 1.38±2.12* |
| | 10 | 1.21±0.16 | 1.5±0.15 | 1.94±0.08[Z] | 1.97±0.08[L] | 1.69±2.05* |
| | 19 | 1.19±0.17 | 1.02±0.14[L] | 1.58±0.06[M] | 1.51±0.08 | 1.48±1.89 |
| Duration in center (s) | -5 | 6.04±0.97 | 7.2±0.97 | 6.5±0.40 | 7.75±0.58 | 6.98±12.6 |
| | -1 | 1.46±0.22 | 1.66±0.36 | 2.05±0.15 | 2.23±0.21 | 2.04±4.54* |
| | 5 | 0.06±0.05[L,Z] | 0.25±0.07[L,Z] | 1.7±0.11[M] | 1.89±0.15[H,M] | 1.54±3.23* |
| | 10 | 1.34±0.19 | 1.65±0.19 | 2.1±0.10 | 1.77±0.13 | 1.89±2.89* |
| | 19 | 1.48±0.25 | 1.27±0.20[L] | 1.99±0.12[M] | 2.11±0.20 | 1.94±3.94 |
| Duration in mirror zone (s) | -1 | 8.24±0.86 | 6.38±0.57 | 6.57±0.25 | 8.18±0.49 | 7.25±9.43 |
| | 5 | 53.8±0.52[M,L,Z] | 42.4±0.80[H,L,M] | 26±0.42[H,M] | 26.4±0.87[H,M] | 29.3±17.7* |
| | 10 | 32.1±1.34[M,L,Z] | 25.3±1.18[H,L] | 19.5±0.45[H,M] | 25.4±0.87[H,L] | 23±15.9* |
| | 19 | 26,3±1.52[L,Z] | 25.9±1.17[L,Z] | 19.7±0.46[H,M,Z] | 21.1±0.66[H,M,L] | 21.1±14.7 |
| Distance moved in mirror zone (cm) | -1 | 46.5±3.00 | 39.4±2.37 | 39.4±1.04 | 31.4±0.96 | 37.2±28.3 |
| | 5 | 234±9.87[M,L,Z] | 211±4.96[H,L,Z] | 116±1.66[H,M,Z] | 47.2±1.15[H,M,L] | 108±75.4* |
| | 10 | 141±6.60[L,Z] | 131±6.60[L,Z] | 98.7±2.29[H,M,Z] | 75.4±2.03[H,L,Z] | 96±65* |
| | 19 | 129±8.11[L,Z] | 134±6.21[L,Z] | 98.2±2.22[H,M,Z] | 77.3±2.09[H,M,L] | 95.9±64.7 |

"duration moving", aggression groups differed in a similar manner, again with group ZA spending the shortest "duration moving", followed by HA, LA and finally MA moving for longest durations (Fig 4B).

The aggression groups differed in "duration in center zone" throughout both tests (LMM, $F_{24,40164}$ = 52.395, p = <0.001) in the order LA > ZA > MA > HA (Table 1, Fig 4C). Over all minutes, HA and MA aggressive groups both spent least time in the center zone (LMM contrast, HA vs. LA: z = -6.706, p<0.001; HA vs. ZA: z = -4.891, p<0.001; HA vs. MA: z = -1.302, p = 1.000; MA vs.LA: z = -5.619, p<0.001; MA vs. ZA: z = -3.649, p<0.001; Fig 4C). Group ZA spent the longest "duration in the center" (LMM contrast, HA vs ZA: z = -4.891, p = <0.001, LA vs ZA: z = 3.249, p = 0.001, MA vs ZA: z = -3.649, p = <0.001). Five minutes into the MIS test, there was a dip in "duration in center" for HA (LMM contrast, HA vs. LA: z = -6.357, p<0.001; HA vs. ZA: z = -5.569, p<0.001) and MA (MA vs. LA: z = -5.982, p<0.001; MA vs. ZA: z = -5.569, p<0.001; Fig 4C). Variable "frequency in center zone" also differed significantly between the aggression groups throughout the tests (LMM, $F_{24,40164}$ = 52.395, p<0.001). Again, the order of the groups was (from high to low frequency in center): LA > ZA > MA > HA (Fig 4D).

## Correlations between activity, boldness and aggression variables

We conducted pairwise correlations between our six variables (two activity, two boldness and two aggression variables), both for all fish and within each aggression group. Variable pairs of activity, boldness and aggression variables showed strong correlations within each pair (Table 2).

There was a positive correlation between the two activity variables and frequency of entries into the center zone (boldness) (Table 2). However, correlation coefficients between activity variables and the other boldness variable, "duration in center zone", were close to zero. This general pattern was also seen in the group-wise correlations, apart from the ZA group, where the correlation between "distance moved" and "duration in center zone" was stronger (robust r = 0.505).

For all aggression groups combined, we found weak negative correlations between activity variables "distance moved" and "duration moving" and aggression variable duration in mirror zone (Table 2). This was true for almost all groups besides HA group, where the correlation between "distance moved in arena" (at min -1) and "distance moved in mirror zone" (at min 5) was moderate (robust r = 0.42).

There was a negative correlation for all aggression groups between boldness variables "frequency in center" (at min -1) and "duration in mirror zone" (at min 5), with a correlation coefficient close to 0. The same variables were correlated for MA and ZA groups however with a correlation coefficient close to 0. There was also a negative correlation between "duration in center zone" (at min-1) and "distance moved in mirror zone" but again with a correlation coefficient that was very close to 0. There was no group-wise correlation for these variables.

## Lateralization

**Head direction while in arena.** For the arenas where the mirror was placed on the left short side, we detected a preference for fish to view the mirror with the left eye (Fig 5A). For these arenas, most fish had an average head direction in the L2 octant while much fewer fish preferred the R2 octant (GLM contrast, z = 4.429, p<0.001). This disbalance was not seen for the L1 vs. R1 octant (GLM contrast, z = -0.806, p = 1) and L3 vs. R3 octant (GLM contrast, z = -0.980, p = 1). However, this preference was not observed for those arenas where the mirror was placed on the right side (Fig 5B). When comparing aggression groups, the HA group more often had a head direction in the R1 octant compared to the LA group (GLM, z = 4.133, p<0.001) and the ZA group (GLM, z = 3.721, p<0.001). Also, this pattern was only found for those parr tested in an arena with the mirror on the left side (Fig 5B).

**Head direction while in mirror zone.** When the fish were in close proximity to the mirror, almost all fish had a head direction in octant L1, L2, R1 or R2, i.e. were oriented parallel to perpendicular towards the mirror (Fig 5C). More fish had a head direction in octant L1 and L2 compared to L3 (GLM contrast, L1 vs. L3: z = 9.397, p<0.001; L2 vs. L3: z = 8.680, p<0.001). Also, octant R1 and R2 were more common than R3 (GLM contrast, R1 vs R3: z = 8.566, p<0.001; R2 vs. R3: z = 7.173 p<0.001). And between L1 and L2 vs R2 there was a significant interest for the L sections (GLM contrast, L1 vs R2: z = 4.802, p<0.001; L2 vs. R2: z = 3.864 = p = 0.003). There was no significant difference between L1 and L2 vs R1.

## Discussion

The aim of the study was to develop a high throughput method for screening behavioral profiles of individual parr, a method that could be applied to identify highly aggressive fish that could be excluded. Even though we could not verify a correlation between aggression and activity or boldness, the results of our study clearly show that the method developed is robust

**Table 2. Pearson's correlation coefficient and robust correlation coefficients between behavioral variables.** Activity and boldness were determined at minute -1, at the end of the OF test when hatch 2 was still in place and the mirror was not visible yet. Aggression variables were determined at minute 5 of the MIS test, when aggression variables peaked.

| Activity variable | Boldness variable | All groups | | | HA | | | MA | | | LA | | | ZA | | |
|---|---|---|---|---|---|---|---|---|---|---|---|---|---|---|---|---|
| | | robust r | Pearson's r | p-value | robust r | Pearson's r | p-value | robust r | Pearson's r | p-value | robust r | Pearson's r | p-value | robust r | Pearson's r | p-value |
| Distance moved in min -1 (cm) | Frequency in center zone at min -1 | 0.40 | 0.38 | **<0.001** | 0.40 | 0.40 | **<0.001** | 0.20 | 0.40 | **<0.001** | 0.21 | 0.31 | **<0.001** | 0.63 | 0.47 | **<0.001** |
| | Duration in center zone at min -1 (s) | 0.22 | -0.05 | **0.033** | 0.10 | 0.10 | 0.415 | -0.10 | 0.20 | **0.038** | 0.03 | -0.13 | **<0.001** | 0.51 | -0.01 | 0.861 |
| Duration moving in min -1 (s) | Frequency in center zone at min -1 | 0.18 | 0.34 | **<0.001** | 0.20 | 0.40 | **<0.001** | 0.10 | 0.20 | **0.033** | 0.01 | 0.26 | **<0.001** | 0.57 | 0.45 | **<0.001** |
| | Duration in center zone at min -1 (s) | 0.07 | -0.04 | 0.157 | 0.00 | 0.20 | 0.093 | -0.10 | 0.10 | 0.541 | -0.08 | -0.11 | **0.002** | 0.51 | 0.02 | 0.606 |
| **Activity variable** | **Aggression variable** | | | | | | | | | | | | | | | |
| Distance moved in min -1 (cm) | Duration in mirror zone at min 5 (s) | -0.19 | -0.18 | **<0.001** | -0.30 | -0.10 | 0.157 | -0.20 | -0.20 | **0.015** | -0.14 | -0.11 | **0.002** | -0.45 | -0.35 | **<0.001** |
| | Distance moved in mirror zone at min 5 (cm) | 0.29 | 0.25 | **<0.001** | 0.40 | 0.40 | **<0.001** | 0.20 | 0.10 | 0.342 | 0.12 | 0.10 | **0.005** | 0.03 | 0.01 | 0.854 |
| Duration moving in min -1 (s) | Duration in mirror zone at min 5 (s) | -0.13 | -0.15 | **<0.001** | 0.20 | -0.10 | 0.255 | 0.10 | 0.00 | 0.885 | -0.17 | -0.05 | 0.121 | -0.50 | -0.34 | **<0.001** |
| | Distance moved in mirror zone at min 5 (cm) | 0.18 | 0.27 | **<0.001** | 0.30 | 0.40 | **<0.001** | 0.10 | 0.10 | 0.238 | 0.05 | 0.10 | **0.005** | 0.01 | 0.03 | 0.547 |
| **Boldness variable** | **Aggression variable** | | | | | | | | | | | | | | | |
| Frequency in center zone at min -1 | Duration in mirror zone at min 5 (s) | -0.11 | -0.10 | **<0.001** | -0.10 | 0.00 | 0.827 | -0.10 | -0.30 | **0.002** | -0.04 | -0.04 | 0.221 | -0.19 | -0.15 | **<0.001** |
| | Distance moved in mirror zone at min 5 (cm) | 0.04 | -0.01 | 0.642 | 0.10 | 0.10 | 0.490 | -0.30 | -0.20 | **0.036** | -0.05 | -0.05 | 0.138 | 0.02 | 0.02 | 0.558 |
| Duration in center zone at min -1 (s) | Duration in mirror zone at min 5 (s) | -0.10 | -0.03 | 0.187 | -0.10 | 0.00 | 0.642 | -0.20 | -0.10 | 0.417 | -0.03 | -0.01 | 0.740 | -0.14 | -0.01 | 0.732 |
| | Distance moved in mirror zone at min 5 (cm) | 0.02 | -0.07 | **0.009** | 0.00 | -0.10 | 0.412 | -0.20 | -0.10 | 0.456 | -0.08 | -0.05 | 0.161 | 0.03 | -0.02 | 0.625 |

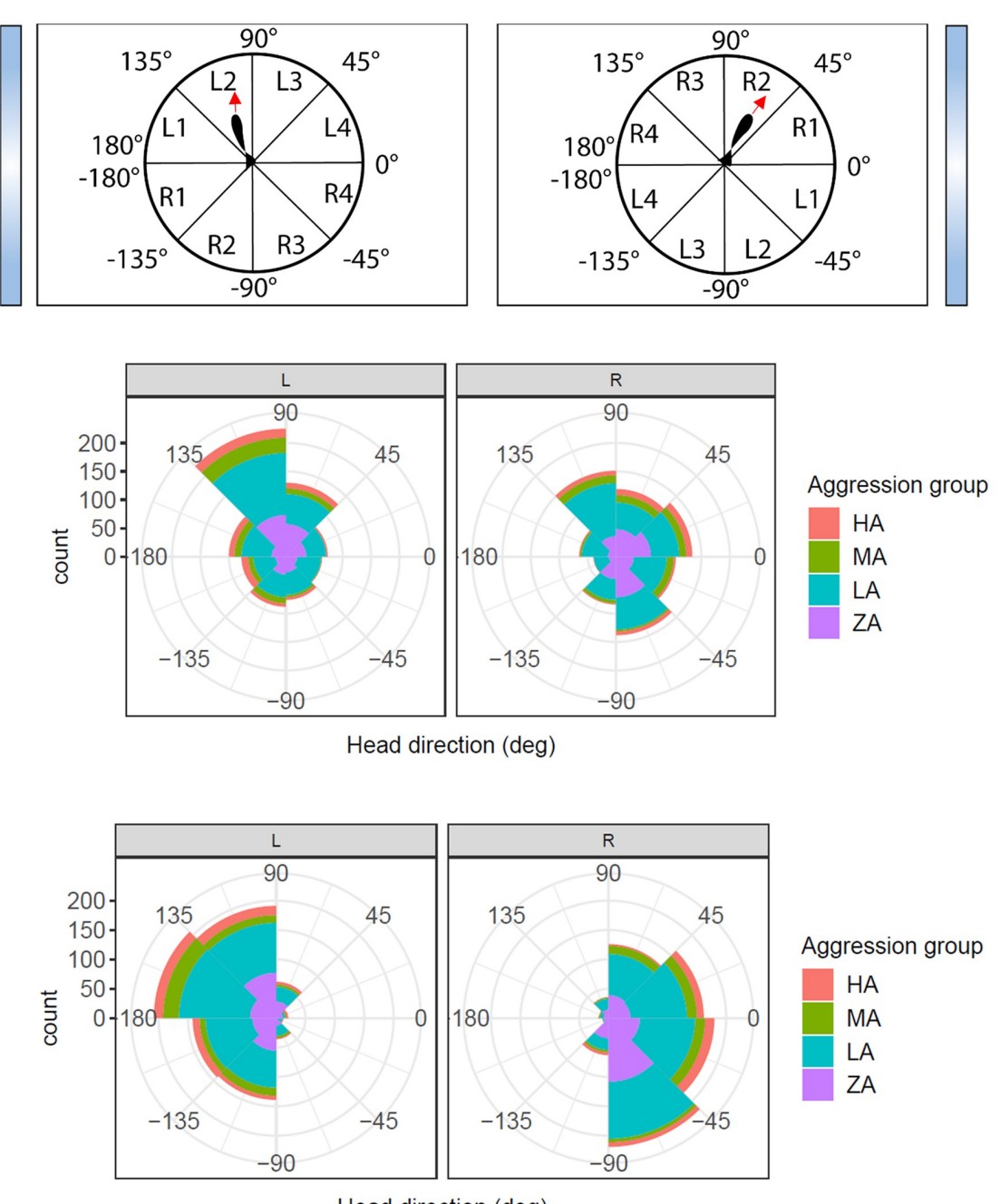

**Fig 5. Head direction during minute 5 of the MIS test.** "L" refers to mirror being place on the left side of the arena and "R" refers to the mirror being on the right side of the arena. A) Examples of how the variable head direction (in degrees, from -180 to 180) was recoded into eight octants (45 degree 'cake pieces'). Left panel: when the fish has a head direction falling into octant L1-L4 this indicates a preference to use the left eye when looking at the mirror. Right panel: a head direction in octant R1-4 indicates a preference to use the right eye. Note that the octant division is equivalent for arenas with the mirror on the left side (left panel) and on the right side (right panel). B) The number of fish per octant in minute 5. The height of the colored rim represents the number of fish per aggression group. C) The same as B), but now the mean head direction was calculated for those instances that the fish was in the mirror zone.

giving valuable information on individual behavioral profiles of salmon parr. Behavioral screening of individual fish is labor-intensive and risk taking at group level [53] and hypoxia avoidance [54] have been suggested as alternative methods for identifying specific behavioral profiles in fish. However, when screening behavior in groups agonistic interactions may confound behavior of individual fish. Moreover, the relationship between hypoxia avoidance, aggression and boldness is not well studied in Atlantic salmon. Therefore, we chose to test salmon individually in two tests that have previously been shown to give a reliable measure of aggression in trout [55].

We found that the fish showed considerable interest in the mirror, as indicated by an increase in duration spent close to the mirror in the first minutes of the MIS test. The head was directed towards the mirror when the fish was inside the mirror zone which is similar to what has been reported for brown trout [55]. By combining both manual scoring ("striking") and automated tracking variables we were able to distinguish separate aggression groups for a large proportion of the tested fish (84%). Automated tracking variables used for allocating fish to different aggression groups were time spent in mirror zone and distance moved in mirror zone during minute five. Still, in many cases these as well as other tracking variables differed between aggression groups throughout the entire test, suggesting that fish of different aggression groups display divergent behavioral profiles. For instance, fish of aggressive groups, MA and LA, all appeared more active than fish of the HA and ZA groups, showing larger distance moved and longer duration moving during both OF and MIS test periods. The LA and ZA fish displayed a less pronounced peak in mirror zone duration and distance moved in mirror zone than HA and MA fish, suggesting lower responsiveness and a more rapid habituation to the mirror stimuli in the LA and ZA fish. For boldness, differences between the groups were less pronounced. In fact, the LA and ZA fish showed higher values for both duration in center zone and frequency in center zone than fish in HA and MA groups. This is an interesting result and opposite to what was expected. In fact, this suggests that low aggressive fish are bolder than high aggressive fish. However, this negative relationship will have to be confirmed by future studies before we can draw any clear conclusions.

Activity, boldness and aggression have been suggested to be correlated forming a behavioral syndrome [39, 41, 56–58]. We observed positive correlations between activity (distance moved and duration moving) and boldness (distance moved in center zone and frequency in center zone). However, we did not find any clear positive correlations between activity and aggression (distance moved in mirror zone and duration in mirror zone) nor between boldness and aggression. This result is not novel, previous studies have suggested that correlations between behavioral traits may differ between populations. In stickleback, bolder males were more aggressive towards other individuals when predation pressure was high, but this correlation was absent when the predation pressure was low [37]. In fact, Bell and Sih [37] showed that if a population of three-spined sticklebacks that does not show any correlation between boldness and aggression, is subjected to predation, such a correlation rapidly develops. Moreover, using MIS tests, Archard and Braithwaite [17] reported that in the poeciliid fish, *Brachyrhaphis episcopi*, individuals from low predation habitats spent more time in front of the mirror than those from high predation habitats.

Clearly, behavioral syndromes such as the correlation between boldness and aggression may be shaped by a combination of genetic constraints and adaptations to environmental conditions. The river Dalälven population of Baltic salmon used in the current study is sea-ranched, a management strategy used to compensate for the loss of natural spawning due to the construction of hydroelectric power plants and dams. Mature male and female fish are caught when ascending and stripped for eggs and milt. Fertilized eggs and later on parr are cultured in a hatchery until smoltification when they are released in the natural environment.

Thus, selective pressures may differ considerably from that of wild populations, especially since sea-ranched fish are not exposed to predation during the parr stage. Moreover, in the hatchery these fish are kept at high densities in a very stable and predictable environment with food in excess. There is now a large body of literature comparing behavioral traits of domesticated and wild populations, which is in most cases consistent in reporting domesticated strains to be bolder and more aggressive on average than their wild counterparts [38, 59–61]. However, when comparing wild masu salmon (*Oncorhynchus masou*) parr with sea-ranched parr there was no difference observed in aggression level neither in the ability to become socially dominant [62]. Brelin et al. [63] studied behavior and physiological stress responses in off-spring of several Swedish brown trout population reared under identical conditions, including the river Dalälven population. They observed that fish displaying what they characterized as a proactive stress coping style were more common in the river Dalälven population, which is heavily affected by sea-ranching, than in populations from smaller rivers less affected by hatchery rearing. Thus, hatchery rearing, and domestication may result in a selection for typical proactive traits, e.g., aggression, boldness, low cortisol but high catecholamine responses to stress. Even though Brelin et al. [63] did not report correlations between aggression and boldness, or between aggression and activity, trout from the Dalälven populations showed higher activity but were significantly less aggressive than fish from populations less affected by sea-ranching. Interestingly, Sadoul et al. (2021) [64] hypothesized that domestication may result in the development of a specific coping style, referred to as preactive and characterized by low aggressiveness and physiological stress responses, but enhanced behavioral plasticity, boldness, and cognitive abilities.

It should be acknowledged that agonistic behavior is difficult to quantify. Both MIS and dyadic fights have their advantages and disadvantages. An agonistic interaction between two (size-matched) fish can be manually scored by an experimenter, either directly or from a video recording [18, 27, 65, 66]. This method can distinguish between many different aspects of the aggressive display but is labor intensive and may raise ethical concerns. In addition, the focal animal may behave differently in response to different opponents, therefore one should ideally test the same individual in several dyads [67]. Fighting a mirror image solves this problem [27, 68]. Nevertheless, the experience of fighting a real opponent is clearly very different compared to fighting one's own mirror image, not in the least since the mirror image will never show any submissive behavior. Agonistic interactions with real opponents include threatening displays, color change or direct attacks with chase and bites [15], all of which helps to acquire information about the ability of the opponent [69]. Behavioral responses towards the mirror reflection and towards a real opponent were correlated in some species of fish [18, 31, 70] but not in others [31]. Using the brain expression of immediate early genes, it has been shown in zebrafish that the brain activation pattern of fish fighting a real opponent differs from that of fish fighting their mirror image [71]. Zebrafish fighting their mirror image also show differences in the activation of the brain monoaminergic systems as compared to zebrafish fighting a real opponent [72]. It may be expected that the motivation to show aggressive behavior is lower in MIS as compared to staged fights with a real opponent. The validity of the MIS test may depend on the morphology and behavior of the species tested, including interspecific differences in lateralization of aggressive behavior [70, 73–75]. Johnsson and Näslund [9, 76] concluded that the MIS test may be better suited for younger fish, as older fish may not register the mirror image as a natural competitor.

Thus, it may be argued that the lack of correlations between activity and aggression as well as between boldness and aggression may be an artifact of the use of the MIS test to quantify aggressive behavior. The time course of the behavioral variables speaks against such an interpretation. In the present study the fish showed a rapid response to the mirror. Interactions

with the mirror image, as indicated with fish being active in the mirror zone, peaked at approximately 5 min after removal of the second hatch in HA and MA fish, whereas it peaked earlier but at a lower level in LA and ZA fish. Following the peak, interactions with the mirror rapidly declined to reach a low and constant level from about 10 min into the MIS test. This suggests that the fish rapidly habituated to, and lost interest in the mirror.

Usually, consistency over time and context are considered perquisite for personality traits like aggression. In the present study behavior was only screened once which is a limitation that may be related to the lack of correlation between behavioral traits. However, in practice it is often difficult to confirm if behavioral traits are consistent over time [77]. In order to avoid habituation, when using a similar arena repeatedly, the intertest interval must be long. However, as a consequence of growth and development, the fish may enter a different life stage, e.g. smoltification or sexual maturation (males). Moreover, it is difficult to keep environmental factors like water temperature constant. Still, Näslund and Johnsson [53] showed that in brown trout fry activity in OF and aggression as determined in a mirror test, similar to the one applied in our study, is repeatable. However, in our study with the large number of fish screened, fish that were older and closer to smoltification alternatively sexual maturation (males), and changing water temperatures, it was more or less impossible to study repeatability in behavioral traits.

Lateralization has been suggested to reflect stress coping styles and to be related to aggression in various vertebrates, including some teleosts (reviewed by Berlinghieri et al. [16]). Berlinghieri et al. [16] introduced the interesting idea that lateralization could potentially be used to select less aggressive fish and by that improve fish welfare in ornamental fish rearing and aquaculture. In our study we could not confirm any eye preference in the MIS test. There was a preference of the fish for using the left eye but only in arenas where the mirror was placed on the left side.

In conclusion, the result of the present study clearly demonstrates that our technique, to combine automatic tracking with manual scoring of a subset of the fish, can be applied for large scale behavioral profiling in juvenile salmon. Our results show different behavioral responses between the aggression groups, where the high aggression group interacted more with the mirror and the medium and low aggression groups were more active. However, we did not observe any clear relationship between boldness and aggression nor between activity and aggression in this population of sea-ranched Baltic salmon. In future studies it would be interesting to include genomics, such as genome-wide association studies (GWAS), which may make it possible to identify genetic markers for aggression.

## Acknowledgments

We are grateful for technical assistance by the staff at Älvkarleby hatchery (Swedish Agricultural University). We would also like to thank Airon Liun and Fredrik Axling for great work effort related to the project. The behavioral testing was carried out by support of the Uppsala University Behavioral Facility (UUBF), Disciplinary Domain of Medicine and Pharmacy, Uppsala University.

## Author Contributions

**Conceptualization:** Svante Winberg.

**Data curation:** Johanna Axling, Laura E. Vossen, Erik Peterson.

**Formal analysis:** Johanna Axling, Laura E. Vossen, Erik Peterson.

**Funding acquisition:** Svante Winberg.

**Investigation:** Johanna Axling.

**Methodology:** Johanna Axling, Laura E. Vossen, Erik Peterson.

**Project administration:** Svante Winberg.

**Resources:** Svante Winberg.

**Supervision:** Erik Peterson, Svante Winberg.

**Writing – original draft:** Johanna Axling.

**Writing – review & editing:** Laura E. Vossen, Erik Peterson, Svante Winberg.

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
