## [Decision Letter · Decision Letter 0]

26 Sep 2022

PONE-D-22-16761Boldness, activity, and aggression: insights from a large-scale study in Baltic salmon (Salmo salar L)PLOS ONE

Dear Prof. Svante Winberg ,

Thank you for submitting your manuscript to PLOS ONE. After careful consideration, we feel that it has merit but does not fully meet PLOS ONE’s publication criteria as it currently stands. Therefore, we invite you to submit a revised version of the manuscript that addresses the points raised during the review process.

In particular the reviewers underline several concerns for example about the consistency of results, the explanation of methods, the link between introduction and discussions. Moreover I agree with the reviewers that the text could be improved in term of clarity. 

We look forward to receiving your revised manuscript.

Kind regards,

Pierluigi Carbonara, PhD

Academic Editor

PLOS ONE

Journal Requirements:

2. In your Methods section, please provide additional information regarding the permits you obtained for the work. Please ensure you have included the full name of the authority that approved the field site access and, if no permits were required, a brief statement explaining why

3. In your Methods section, please include a comment about the state of the animals following this research. Were they euthanized or housed for use in further research? If any animals were sacrificed by the authors, please include the method of euthanasia and describe any efforts that were undertaken to reduce animal suffering.

Reviewers' comments:

Reviewer's Responses to Questions

**Comments to the Author**

1. Is the manuscript technically sound, and do the data support the conclusions?

Reviewer #1: Yes

Reviewer #2: Partly

Reviewer #3: Yes

2. Has the statistical analysis been performed appropriately and rigorously? 

Reviewer #1: Yes

Reviewer #2: Yes

Reviewer #3: Yes

3. Have the authors made all data underlying the findings in their manuscript fully available?

Reviewer #1: Yes

Reviewer #2: No

Reviewer #3: Yes

4. Is the manuscript presented in an intelligible fashion and written in standard English?

Reviewer #1: Yes

Reviewer #2: Yes

Reviewer #3: Yes

5. Review Comments to the Author

Reviewer #1: PONE-D-22-16761 :

General comment :

The paper by Axling et al. evaluates the link between boldness, activity and aggression in Baltic salmon using open field and mirror image simulation (MIS) tests using large sample size (~2000 parr). Overall, they found that activity and boldness were positively correlated, and that low and medium aggressive fish were more activity than highly aggressive ones. Also, they conclude that, by combining automated and manual scoring, the MIS is relevant for large-scale aggression profiling in salmons.

The manuscript is well written with clear description of M&M (except one point discussed below). The statistical analysis is sounds and was well conducted by the authors. Also, I would like to congratulate the authors for the huge amount of work carried out here by testing almost 2000 salmons.

There are however some major points that need to be discussed by the authors before publication of the manuscript.

1) Typically, boldness, activity and aggression are considered as personality traits and should be measured in several time point to assess consistency of behavioural response, which has not been performed here. Does the response in OF and MIS is consistent over time ? (checked in previous publication ?) If not, this is a limitation point of the study, that could explain the lack or inverse of the correlation expected (see specific comments). This should be introduced and discussed by the authors

2) Discussion of some intriguing points regarding the fish behavioural response during test over time are overlooked by the authors and should be discussed (e.g. time spent in center zone decrease over time while the opposite is generally observed in open field test; see specific points).

3) L188, the authors explain the assignation criteria to assign fish to one or other groups (from 0 sec aggression (ZA) to 50/60 sec/min of striking (HA)) and in the results (L270), the authors wrote: "In total 1679 out of 1987 fish (84%) could be assigned to an aggression group". It is unclear why 16 % of fish were not assigned to one of the groups because the criteria of assignation should find a group to these fish. Also, I don't understand why the verb form "could" is employed. This could be due to the fact only 5 % of fish were manually scored for striking and the method used for assignation of the 95 % remaining (L194-196) or I maybe miss something else but it is unclear for me. Please better argue and explain this point to the readers.

4) This is huge amount of work to screen individually fish to get aggressive profile (also by using proxy of aggression which were expected to be boldness/activity). Why the authors did not tried to correlate aggression with more high throughput tests such as group risk-taking test and/or hypoxia tests ? This is common in teleost, such as in zebrafish, European sea bass or sea bream. Also, there is some paper in Salmon (e.g. Damsgård et al. 2019/ https://doi.org/10.1098/rsos.181859). I believe this point deserves some discussion due to research objectives of the present paper.

Specific comments :

L18 : suggest to replace ‘traits from other context’ by ‘other behavioural traits’

L57: coma after conspecific

L62: not clear

L66: Please define personality trait (see general comment 1).

L67: same comment than in the abstract

L70: locomotor

L73-75: Please be more specific ? how it was questioned ?

L87: why OF has been choosed ? this should be argued.

Some other methods have been developed for similar purposes in salmon (e.g. Damsgård et al. 2019/ https://doi.org/10.1098/rsos.181859; see general comment 4). This could be confronted in the discussion section with the method developed in the present study.

L126: please provide mean +/- SD for tested fish.

Statistical section: Do the authors check the assumptions (e.g. normality of residuals, independence, etc.) of GLMMs they did ? This has to be checked. One sentence such as “Visual inspection of the residuals revealed no violation of the statistical assumptions by the model” should be added to the section to ensure that the authors ensured that.

Also, it appears that mass/size of fish has not been include in the model. Does it affect behavioral response ? please argue on that point.

L270 : It is not clear why 16 % of fish could not be assigned to an aggression group. What was the reason ? too much aggressive for the high aggressive group ? Please better explain and discuss that point.

L383: It was 84 L270. Please discuss the point highlighted in L270.

L409: please add a coma after “tests”

L420-429: I would suggest to the authors this paper regarding effects of human protection, including domestication, on coping styles by Sadoul et al. 2021; https://doi.org/10.1371/journal.pbio.3001186

L431: “did not report correlations […] appeared less aggressive”. What do the authors mean ? Is that a personal communication/observation? please add more details to argue this point.

L459: There is a need for more discussion regarding the results of the OF test since the authors did not discuss at all what happening during this test. Especially in my opinion, the fact that progressively fish spend less time in the center zone over time appears unexpected. Typically, fish are progressively exploring more the center zone over time, since this area is riskier that periphery. Some recent examples in farmed species such as European sea bass advocate for that (e.g. Alfonso et al. 2020; https://doi.org/10.1016/j.applanim.2020.104974 / Sadoul et al. 2022 ; https://doi.org/10.1098/rspb.2022.0399) but this is also the case for other species, isn’t it ? What about salmons ? This needs to be deeper discussed since appears counter intuitive.

Also one major point that it is not discusses by the authors is that the lack of correlation between boldness, activity and aggression could be explained by the fact that the authors only did one test to assess these variables (OF and MIS) performed on the same moment. Typically, boldness, activity and aggression are overall considered as personality traits, and should be consistent over time. This has not be ensured by the authors. Do the authors have evidence from previous publications that the behavioral variables recorded are consistent over time ? Please discuss that point in the discussion. This is, in my opinion, a limitation point that need further discussion.

Figures:

Figure 2: It is not clear enough what the dimensions are referring for. Please be clearer by adding arrows.

Figure 3: Maybe could be move in matt supp ?

It could be nice to see the correlations between the different behavioural traits (shown in table 2) as a figure. Maybe for the material supp ?

Reviewer #2: This study explores a variety of behavioral traits in a Baltic population of Atlantic salmon via tests in a laboratory set up. Specifically, the authors explore the relationship between aggressive behavior and other behavioral measurements including activity, boldness and lateralization. Results are interesting and the authors highlight the potential importance and implication of the results of this study in aquaculture and specifically salmon farms. Despite the potential of the study I think some revisions should be made in order to clarify the methods, analyses and make easier the understanding and flow of the manuscript. In addition, a link between introduction and discussions is missing, I would have expected more discussion about the implication of the results and aquaculture farms management. Please see specific comments about the manuscript below.

There is some inconsistency in the order of description of the methods, such as measurements of behavioral traits, correlation among those variables and effects on groups with different aggression levels. For example, in the abstract authors first introduce the correlations, then the division in groups with different aggression levels and finish by mentioning the lateralization tests. However, in the Statistical analyses session of the methods authors first describe the division in aggression groups etc. A re-organization of the order would make a lot clearer the logical and structure of the paper.

Abstract: the discussion/conclusion could be extended, not clear why the authors conclude that OF may not accurately predict aggressive behavior (Line 33) while in the previous sentence it is written that activity varies across groups with different aggression levels (line 30).

Introduction:

The introduction could be a bit more focused, for example, in the first paragraph the consequences of aggression in aquaculture (fin damage, infections, ecc..) can be shortened in one sentence and leave some space to introduce a bit more the purpose of the paper such as the importance of the link between aggression and other behavioral traits.

L69: In the same species or in other species?

L75: Missing a link with the next paragraph.

L76-83: This paragraph could be extended by introducing more generally the methods of the laboratory set up for determining fish behavior, highlighting the relevance of the laboratory test used in this study

L84-96: At the end of the introduction, hypothesis and predictions could be mentioned.

Methods:

L121: Average size of fish?

L130-131: Could move this sentence to the end of the paragraph and try to avoid repetitions.

L132: Should explain what OF and MIT stands for.

L138: In the figure legend it should be mentioned what OF stands for.

L155: Could mention that those analyses are not included in the manuscript, otherwise we expect to see some results about genomic analyses.

L158: Change “gram” to “g”

L184: Why most reliable?

L187-188: Not clear how the aggression is measured in striking seconds per minute? In the figure 3 legend it much clearer: “number of seconds spent displaying striking behavior”.

L188-189: On what the division into groups was based on? Why you didn’t use the extreme of 0.01-10 sec/min for the low aggression group?

L210: I suggest to change “differed in behavior” with “had an effect on all behavioral trait measured”

L210: Are those models referred to the effect of group behavior? Or are those models fitted to explore the effect of time of the trial on the behavior as shown in the first paragraph of the results and in figure 4? If that is the case, then it seems that the description of the model exploring the effect of group aggression on behavior is missing.

L210-211: were response variable measured as behavior per minute, such as distance moved in the mirror zone per minute? I would suggest to mention it here as well. It would make clearer to understand why fish ID was used as a random factor.

L219: Why temperature was removed from this model?

L222: And random factors?

L224-230: You should mention if those correlations were measured for all groups or just within aggression groups, from the results it sounds it was done in both overall groups and within groups but from the methods this information is missing.

L233-L235: Was this model done using data from fish recorded in the whole arena?

L270: what happened to the other 16%? Why couldn’t they be assigned to any group?

Discussion:

Overall, the discussions and conclusions could be more focused on the importance of the results within the context of aquaculture farms, as presented in the introduction. I think a link between introduction and discussions is a bit missing.

L398: I would suggest to start the paragraph showing the results that have been found in the present study and then discuss if those results fit the predictions or not and why yes/not. It would make clearer to follow the rational.

L414: Could explain what sea-ranched refers to since some readers may not be familiar with the terminology.

L444: Could it be that they are displaying some social behavior instead of aggression?

Figure 1: Remove 0 in the 0-5min start box otherwise need to say 5-10 etc. for the other timelines. Also, what the numbers in brackets refers to? Shouldn’t it be “-5 to 0” for the open field and “0 to 20” for the mirror?

Figure 2: Can you add a line with arrows that indicates the start and end of the arena measurements? As it is now it’s not clear to which exact part of the arena 42cm refers to.

Figure 4: Is this the distance moved per minute? Please make it clearer in the figure as well as in methods and results.

Figure 5: Panel labels are missing which make very difficult to understand which part of the figure results refers to.

Reviewer #3: This manuscript seeks to develop a high-throughput method to screen aggressive behavioural traits in Baltic salmon because of the deleterious effects these fish can have in Aquaculture settings. Characterising aggressive traits can be time-consuming and labour intensive and so the authors investigate whether other easily measured traits, such as boldness and activity, can be used as a proxy for aggression. The authors use open field and mirror imaging tests and assign nearly 2000 fish individual activity, boldness and lateralization scores. On a subset of ~100 fish, they manually assign aggression scores (based on mirror strikes) and from this, they successfully extrapolate aggression scores to the remaining fish. They find no clear relationship between boldness/activity and aggression and head lateralization and aggression but do find a relationship between boldness and activity.

I find the overarching aims and findings of the study to be really interesting and the methods and sample sizes to be robust. I therefore firmly believe the study will make a great contribution to the literature. However, in its current form, I feel as if readers will be lost during the results and discussion sections and the findings and story of the manuscript will be overlooked. I understand the difficulties in displaying lengthy results but I encourage the authors to significantly reduce the text of the results and make their key findings a little clearer. For example, perhaps a portion of this could be done by using tables to display estimated marginal mean comparisons. The discussion also needs slight editing and trimming (i.e. lengthy paragraphs) as I feel it loses focus of the study aims and the key findings. Apart from that, I commend the authors on a great study and provide a few more comments/concerns below. I hope the authors find my suggestions constructive in improving their manuscript.

Introduction

A well written and clear introduction – really nice!

Methods

Lines 126-137: Include how many arenas you had somewhere here.

Line 130: Remove “from the river Daläven” as you mention this in line 136.

Line 137: Include that you recorded temperature for each trial so the reader is less concerned about the large difference in temps.

Line 144: Does this mean there were 4 fish in each arena? Just needs to be made a little clearer.

Line 157: Perhaps provide how many trials were performed as well.

Line 185: Can remove the definition of striking here as you mention it not far above.

Line 187: maybe include “Striking for 50-60 sec/min” here.

Fig 3: Provide y-axis lines and in the caption for A) mention that the colour scale was for the subset of fish manually scored and the grey dots represents all fish from the study.

Line 207: It would be beneficial to include these model results in a supplementary file.

Line 219: Any reason why a random effect of temperature was not used here?

Line 223: How did you assess the significance of factors? Through the model summary or did you run an anova on the model? i.e. code in R would be anova(mod1) to assess the significance of main factors.

Line 226: This is a little unclear. Does this mean correlations were just for the data from the 1st minute and 5th minute?

Results

I encourage the authors to make their data publically available on a platform like figshare.

Line 245 and throughout: When mentioning each variable like “The activity variable duration moving in arena…” perhaps put ‘duration moving in arena’ in quotation marks to help with the ease of reading.

Line 251: The panel “G” is missing in Fig 4.

Line 270/Fig 4: I suggest you provide small subheadings above each panel/ pair of panels to indicate what these represent (i.e. “Activity” or “Boldness”). It might also be worthwhile including boxplots of overall average scores for each aggression level next to these plots.

Lines 331-332: Maybe remind the reader that ‘frequency of entries into the center zone’ refers to boldness.

Lines 348-376: I think you mean “GLMM” here and not “LMM”.

Lines 348-376: Can the results of the LMM contrasts not go into a table?

Discussion

Line 378-397: I suggest you first refer back to the study aims and then continue with key findings. Although there was no relationship found between aggression and boldness/activity, the study methods are robust and the findings are useful so this needs to be driven home in the opening paragraph here.

Line 414: Maybe break this up into two paragraphs.

Line 455-456: Your finding of a strong relationship between boldness and activity is really interesting, especially with respect to wild passive fisheries targeting bold and active phenotypes. Perhaps a paragraph here on this would broaden the audience of your manuscript.

Line 464: Remove ‘and’ from …”and to,…”

Line 479: Perhaps provide a few guidelines for future research.

Overall, the discussion is lacking a strong link with the aims of the study and may therefore lose the readers interest and understanding.

6. PLOS authors have the option to publish the peer review history of their article (what does this mean?). If published, this will include your full peer review and any attached files.

Reviewer #1: No

Reviewer #2: No

Reviewer #3: No

---

## [Author Response · Author response to Decision Letter 0]

6 Feb 2023

Reviewer #1

General comment :

The paper by Axling et al. evaluates the link between boldness, activity and aggression in Baltic salmon using open field and mirror image simulation (MIS) tests using large sample size (~2000 parr). Overall, they found that activity and boldness were positively correlated, and that low and medium aggressive fish were more activity than highly aggressive ones. Also, they conclude that, by combining automated and manual scoring, the MIS is relevant for large-scale aggression profiling in salmons.

The manuscript is well written with clear description of M&M (except one point discussed below). The statistical analysis is sounds and was well conducted by the authors. Also, I would like to congratulate the authors for the huge amount of work carried out here by testing almost 2000 salmons.

Reply: The authors thank Reviewer 1 for investing considerable time and effort in providing us with constructive and helpful comments, which we feel have improved the quality of the manuscript.

There are however some major points that need to be discussed by the authors before publication of the manuscript.

1) Typically, boldness, activity and aggression are considered as personality traits and should be measured in several time point to assess consistency of behavioural response, which has not been performed here. Does the response in OF and MIS is consistent over time? (checked in previous publication ?) If not, this is a limitation point of the study, that could explain the lack or inverse of the correlation expected (see specific comments). This should be introduced and discussed by the authors

Reply: The reviewer raises a valid point since consistency over time has been considered as a prerequisite of personality traits. However, in practice it is often difficult to confirm if behavioral traits are consistent over time. When repeating a behavioral assay, using a similar arena, the arena and the test situation is no longer novel to the fish. Moreover, as a consequence of growth and development, the fish may enter a different life stage, e.g. smoltification or sexual maturation (males). Also, factors like water temperature will change which may have drastic effects on behavioral responses. Still, Näslund and Johnsson (2016) showed that brown trout fry activity in OF and aggression as determined in a mirror test, similar to the one applied in our study, is repeatable. However, in our study with the large number of fish screened, fish that were older and closer to smoltification alternatively sexual maturation (males), and changing water temperatures, it was more or less impossible to study repeatability in behavioral traits. This has now been clearly acknowledged (l. 495-506).

2) Discussion of some intriguing points regarding the fish behavioural response during test over time are overlooked by the authors and should be discussed (e.g. time spent in center zone decrease over time while the opposite is generally observed in open field test; see specific points).

Reply: Thank you, we respond to this comment in the specific comments below.

3) L188, the authors explain the assignation criteria to assign fish to one or other groups (from 0 sec aggression (ZA) to 50/60 sec/min of striking (HA)) and in the results (L270), the authors wrote: "In total ". It is unclear why 16 % of fish were not assigned to one of the groups because the criteria of assignation should find a group to these fish. Also, I don't understand why the verb form "could" is employed. This could be due to the fact only 5 % of fish were manually scored for striking and the method used for assignation of the 95 % remaining (L194-196) or I maybe miss something else but it is unclear for me. Please better argue and explain this point to the readers.

Reply: In order to assign the remaining 95% of the fish two video tracking variables (the duration of time spent in the mirror zone and the distance moved in the mirror zone) were used; if one or both of these variables had missing values an individual could not be assigned. The result was that 16% of the fish could not be assigned to an aggression group. Thus, it has to do with the properties of the specific discriminant analysis used, PROC DISCRIM in SAS. (l. 211-21 5)

4) This is huge amount of work to screen individually fish to get aggressive profile (also by using proxy of aggression which were expected to be boldness/activity). Why the authors did not tried to correlate aggression with more high throughput tests such as group risk-taking test and/or hypoxia tests ? This is common in teleost, such as in zebrafish, European sea bass or sea bream. Also, there is some paper in Salmon (e.g. Damsgård et al. 2019/ https://doi.org/10.1098/rsos.181859). I believe this point deserves some discussion due to research objectives of the present paper. 

Reply: We thank the reviewer for this important and insightful comment. We chose to use two individual tests because in group tests for boldness, social and/or agonistic interactions may confound individual responses. Furthermore, there is little research on salmon on how the results of group-based or hypoxia test correspond to individual tests of boldness or aggression. We were interested in the question if and how activity and boldness are related to aggression at an individual level, and ultimately, what the underlying genetic and neuroendocrine differences are between salmon of different behavioral types. We have added this to the discussion section at line 399-406.

Specific comments :

L18 : suggest to replace ‘traits from other context’ by ‘other behavioural traits’

Reply: Thank you, we have changed this. 

L57: coma after conspecific 

Reply: Thank you, we have changed this.

L62: not clear

Reply:

L66: Please define personality trait 

Reply: Thank you, we have added a definition at line 61-70.

L67: same comment than in the abstract

Reply: sorry but we do not understand this comment.

L70: locomotor

Reply: Thank you we have changed this.

L73-75: Please be more specific? how it was questioned?

Reply: This sentence has been deleted.

L87: why OF has been choosed? this should be argued.

Reply: the OF test is a widely used behavioral test and our results can therefore easily be compared with OF results from Atlantic salmon and other fish species. In addition, two of the behaviors of interest in this study, activity and boldness, can be measured in the OF, which is explained two sentences later. 

We have added this explanation to line 85-87 [such as boldness and locomotory activity] that can be measured in the widely used open field (OF) test”.

Some other methods have been developed for similar purposes in salmon (e.g. Damsgård et al. 2019/ https://doi.org/10.1098/rsos.181859; see general comment 4). This could be confronted in the discussion section with the method developed in the present study.

Reply: see our reply to general comment 4.

L126: please provide mean +/- SD for tested fish.

Reply: Thank you we have now added this on line 152-154.

Statistical section: Do the authors check the assumptions (e.g. normality of residuals, independence, etc.) of GLMMs they did ? This has to be checked. One sentence such as “Visual inspection of the residuals revealed no violation of the statistical assumptions by the model” should be added to the section to ensure that the authors ensured that.

Reply: Thank you for bringing this up. We have now added a sentence to the statistical methods section at line 244-246: “For all models, we evaluated assumptions (normality, homoscedasticity, outliers) using residual plots. For the negative binomial GLMM we also assessed overdispersion.“

Also, it appears that mass/size of fish has not been include in the model. Does it affect behavioral response? please argue on that point.

Reply: The reason why we did not include weight as a covariate in our model is because we were interested in absolute activity, boldness and aggression levels and not weight-adjusted levels. We were not interested in the causal effect of weight on these variables. However, we did want to adjust for differences due to water temperature and arena, therefore these were added as random effects.

Indeed it is well established that larger fish often win over smaller fish in pair-wise contests. Therefore, individuals are often matched for weight when experimenters can assign pairs, in which case the effect of weight is not always significant. In the mirror stimulation test weight is in a sense already matched since the individual meets a contestor of exactly the same weight. 

L270 : It is not clear why 16 % of fish could not be assigned to an aggression group. What was the reason? too much aggressive for the high aggressive group? Please better explain and discuss that point.

Reply: Thank you please see general comment 3.

L383: It was 84 L270. Please discuss the point highlighted in L270.

Reply: Thank you, we have changed it to 84%. See general comment 3.

L409: please add a comma after “tests”

Reply: Thank you we have changed this.

L420-429: I would suggest to the authors this paper regarding effects of human protection, including domestication, on coping styles by Sadoul et al. 2021; https://doi.org/10.1371/journal.pbio.3001186

Reply: Thank you for your advice we have added a sentence about this on line 465-468.

L431: “did not report correlations […] appeared less aggressive”. What do the authors mean ? Is that a personal communication/observation? please add more details to argue this point.

Reply: Brelin et al. (2008) did not detect any correlations between aggression and boldness, or between aggression and activity, even though trout from river Dalälven showed significantly higher activity (population mean) but significantly lower aggression (population mean) than fish from populations less affected by sea-ranching (line 463-464: “trout from the Dalälven populations showed higher activity but were significantly less aggressive than fish from populations less affected by sea-ranching”). 

L459: There is a need for more discussion regarding the results of the OF test since the authors did not discuss at all what happening during this test. Especially in my opinion, the fact that progressively fish spend less time in the center zone over time appears unexpected. Typically, fish are progressively exploring more the center zone over time, since this area is riskier that periphery. Some recent examples in farmed species such as European sea bass advocate for that (e.g. Alfonso et al. 2020; https://doi.org/10.1016/j.applanim.2020.104974 / Sadoul et al. 2022 ; https://doi.org/10.1098/rspb.2022.0399) but this is also the case for other species, isn’t it ? What about salmons ? This needs to be deeper discussed since appears counter intuitive.

Reply: we thank the reviewer for this insightful comment. Indeed anxiety-related behaviors such as thigmotaxis (duration in wall zone) in the OF test and bottom dwelling in the diving test often decrease over testing time. However, in our OF test duration in center zone was only assessed for 5 minutes, and it is not unlikely that the salmon needed a longer acclimation period before exploring the arena. This initial ‘freezing’ period in the beginning of a test has been shown to be rather species specific and is also heavily affected by handling experience. During the MIS test, a longer time spent near the mirror directly reduced the duration in center. Indeed it seems that the dip in duration and frequency in the center (Fig 4C and D) coincides with a peak in time spent near the mirror (Fig 4E and F). However, after this initial interest in the mirror, the Baltic salmon appear to start exploring the rest of the arena as indicated by an increase in duration and frequency in the center (Fig 4E and F).

Also one major point that it is not discusses by the authors is that the lack of correlation between boldness, activity and aggression could be explained by the fact that the authors only did one test to assess these variables (OF and MIS) performed on the same moment. 

Reply: The reviewer suggests that the lack of correlation between boldness, activity and aggression was explained by our test procedure. We would like to clarify that for the correlations, the level of boldness (duration and frequency in center zone) was measured during the OF test (more specifically minute -1), when hatch 2 was still in place and the mirror was not visible yet. Boldness was therefore measured independently from mirror aggression. However, we realize that reporting the boldness variables also during the MIS test may cause confusion. We have therefore added a sentence to the caption of Table 1 to prevent this wrong interpretation: “Activity and boldness were determined at minute -1, at the end of the OF test when hatch 2 was still in place and the mirror was not visible yet. Aggression variables were determined at minute 5 of the MIS test, when aggression variables peaked.”

Typically, boldness, activity and aggression are overall considered as personality traits, and should be consistent over time. This has not be ensured by the authors. Do the authors have evidence from previous publications that the behavioral variables recorded are consistent over time ? Please discuss that point in the discussion. This is, in my opinion, a limitation point that need further discussion.

Reply: see reply to general comment 1.

Figures:

Figure 2: It is not clear enough what the dimensions are referring for. Please be clearer by adding arrows.

Reply: Thank you for you feedback, we now added this information to the figure. 

Figure 3: Maybe could be move in matt supp ?

Reply: Thank you but we prefer to keep Fig 3 in the text. 

It could be nice to see the correlations between the different behavioural traits (shown in table 2) as a figure. Maybe for the material supp ?

Reply: Thank you for this suggestion. We will upload the raw data to fig share, so it will be possible for readers to investigate many possible correlations. 

 

Reviewer #2

This study explores a variety of behavioral traits in a Baltic population of Atlantic salmon via tests in a laboratory set up. Specifically, the authors explore the relationship between aggressive behavior and other behavioral measurements including activity, boldness and lateralization. Results are interesting and the authors highlight the potential importance and implication of the results of this study in aquaculture and specifically salmon farms. Despite the potential of the study I think some revisions should be made in order to clarify the methods, analyses and make easier the understanding and flow of the manuscript. In addition, a link between introduction and discussions is missing, I would have expected more discussion about the implication of the results and aquaculture farms management. Please see specific comments about the manuscript below.

Reply: We thank the reviewer for his/ her commitment and for providing constructive and insightful feedback. We reply to each comment below, describing the changes made, which we feel have greatly improved the manuscript. 

There is some inconsistency in the order of description of the methods, such as measurements of behavioral traits, correlation among those variables and effects on groups with different aggression levels. For example, in the abstract authors first introduce the correlations, then the division in groups with different aggression levels and finish by mentioning the lateralization tests. However, in the Statistical analyses session of the methods authors first describe the division in aggression groups etc. A re-organization of the order would make a lot clearer the logical and structure of the paper.

Reply: We agree with the reviewer and have adapted the order of the measurements in the statistical analysis section of the methods. We now start with the division into groups, then explain the correlations and finish with the analysis of the lateralization.

Abstract: the discussion/conclusion could be extended, not clear why the authors conclude that OF may not accurately predict aggressive behavior (Line 33) while in the previous sentence it is written that activity varies across groups with different aggression levels (line 30).

Reply: Thank you for this feedback, we have adapted the last sentence of the abstract (line 32-34): “We conclude that aggressiveness cannot be predicted from the results of the OF test alone but that the MIS test can be used for large-scale individual aggression profiling of juvenile salmon.”

Introduction:

The introduction could be a bit more focused, for example, in the first paragraph the consequences of aggression in aquaculture (fin damage, infections, ecc..) can be shortened in one sentence and leave some space to introduce a bit more the purpose of the paper such as the importance of the link between aggression and other behavioral traits.

Reply: Thank you for your comment. We feel however that the aquaculture section adds context to the benefits of large-scale behavioural testing. Our paper is in a sense more applied than many other papers on the link between aggression and other traits. Instead, we refer the reader to already published reviews on the subject.

L69: In the same species or in other species?

Reply: reference 27-29 refer to three-spined sticklebacks. We have adapted the text, see revised manuscript line 88-90.

L75: Missing a link with the next paragraph.

Reply: thank you for bringing this up. We have changed the order of these paragraphs in the introduction. Now we discuss the MIS test and then we discuss that aggression may be correlated with other traits and hence may be measured indirectly. This new order enhances the red line in the introduction. Also, we added a linking sentence (see next comment).

L76-83: This paragraph could be extended by introducing more generally the methods of the laboratory set up for determining fish behavior, highlighting the relevance of the laboratory test used in this study

Reply: we did not want to make this paragraph about the MIS test too long, since we also discuss the MIS test again in the discussion section. However, we added a sentence in line 73-74: “In fish research, behavioral test of aggression exposing the individual to a mirror have gained in popularity, since such tests reduce the need for repeated testing against multiple opponents (REF).”.

L84-96: At the end of the introduction, hypothesis and predictions could be mentioned.

Reply: We did not mention the hypothesis and predictions in the introduction because this is not mentioned in the author guidelines for Plos One. 

Methods:

L121: Average size of fish?

Reply : Thank you for your comment, fork length has been added at line 152-154. 

L130-131: Could move this sentence to the end of the paragraph and try to avoid repetitions.

Reply: at line 144, we replaced the text “filled with water from the river Dalälven to a water depth of 10 cm” with “filled to a water depth of 10 cm”.

L132: Should explain what OF and MIT stands for.

Reply: we have followed the convention that abbreviations are introduced only once (OF on line 87 and MIS on line 75) and thereafter the abbreviations were used. We have made an exception for the figure and table captions.

L138: In the figure legend it should be mentioned what OF stands for.

Reply: we added ”open field” before “OF” in line 156.

L155: Could mention that those analyses are not included in the manuscript, otherwise we expect to see some results about genomic analyses.

Reply:Thank you, we added ”potential” before “genomic analyses” in line 172.

L158: Change “gram” to “g”

Reply: Thank you, we changed “gram” to “g” in line 175.

L184: Why most reliable?

Reply: there are often multiple measures taken in behavioral tests of aggression, but striking is the most escalated behavior, is the most energetically demanding and also has the largest consequences for the loser. It can therefore be considered as a more “honest signal” than for instance a threat display. We added references to line 201 to corroborate this.

L187-188: Not clear how the aggression is measured in striking seconds per minute? In the figure 3 legend it much clearer: “number of seconds spent displaying striking behavior”.

Reply: we adapted the description of striking behavior in line 198: “ Our most reliable measure of aggressiveness was the manually scored variable “striking”, the number of seconds per minute that the individual spent moving perpendicular towards the mirror at a fast speed [9,14,51] .”

L188-189: On what the division into groups was based on? Why you didn’t use the extreme of 0.01-10 sec/min for the low aggression group?

Reply: thank you for this insightful comment. Indeed we did consider the approach suggested by the reviewer at an earlier stage in the analysis. As the reviewer points out, from figure 3A one can see that the duration striking 0.1-10 sec (the blue dots) corresponds to 0-100 cm distance moved in mirror zone. Hence, to define the low aggression group one could have simply taken all the fish (including those for which manual scoring was lacking) that had a distance moved in mirror zone of 0-100 cm. The problem is that this approach did not work for the other aggression groups. The green and red dots in figure 3A are found at various levels of the x and y and the relationship is also not linear (if this would have been the case we could have used the residuals from a linear regression). For this reason we used a discriminant analysis which classified the grey points in figure 3A (for which manual scoring was not available) according to the ‘closest’ colored point (for which manual scoring was available). Closest here means that squared Mahalanobis distance were used to determine proximity.

L210: I suggest to change “differed in behavior” with “had an effect on all behavioral trait measured”

Reply: we changed ”differed in behavior” to ”affected each of the six behavioral variables measured”.

L210: Are those models referred to the effect of group behavior? Or are those models fitted to explore the effect of time of the trial on the behavior as shown in the first paragraph of the results and in figure 4? If that is the case, then it seems that the description of the model exploring the effect of group aggression on behavior is missing.

Reply: We only used one set of 6 mixed effect models. These models investigated the effect of aggression group, but a fixed effect of time was added to adjust for the fact that the response variables varied over time. Also, post-hoc tests on the effect of time (differences between minutes regardless of aggression group) furthermore allowed us to quantify how behavioral variables changed over time.

L210-211: were response variable measured as behavior per minute, such as distance moved in the mirror zone per minute? I would suggest to mention it here as well. It would make clearer to understand why fish ID was used as a random factor.

Reply: Thank you , we added ”per minute” to line 234-235 to clarify.

L219: Why temperature was removed from this model?

Reply: we decided to not include temperature after extensive visual exploration of the data. In contrast to locomotory variables distance moved in arena, proportion of time moving in arena and distance moved in mirror zone, there was no clear effect of temperature on the proportion of time spent in the center zone or mirror zone. The effect of temperature on locomotory behavior of fish is also well established in the literature [54](line 241-243).

L222: And random factors?

Reply: we used the term ‘explanatory variables’ to include both fixed and random effects.

L224-230: You should mention if those correlations were measured for all groups or just within aggression groups, from the results it sounds it was done in both overall groups and within groups but from the methods this information is missing.

Reply: Thank you for pointing this out. We correlated the 6 variables both per aggression group and also derived overall correlation. We added a row for the overall correlations to Table 2.

L233-L235: Was this model done using data from fish recorded in the whole arena?

Reply: No, for the lateralization head direction was measured while the fish was in the mirror zone. We added this information to line 255. Thank you for noticing this!

L270: what happened to the other 16%? Why couldn’t they be assigned to any group?

Reply, Please see our reply to general comment #3 reviewer 1. 

Discussion:

Overall, the discussions and conclusions could be more focused on the importance of the results within the context of aquaculture farms, as presented in the introduction. I think a link between introduction and discussions is a bit missing.

L398: I would suggest to start the paragraph showing the results that have been found in the present study and then discuss if those results fit the predictions or not and why yes/not. It would make clearer to follow the rational.

Reply: Thanks for the comment, we have now added a clarifying section to the discussion on line (395-406, also a response to reviewer 3). 

L414: Could explain what sea-ranched refers to since some readers may not be familiar with the terminology.

Reply: Thank you for your comment. A clarification has been added on line 444-448.

L444: Could it be that they are displaying some social behavior instead of aggression?

Reply: Thank you for your comment and a certain level of interests to the mirror could certainly be display of social interaction however considering the parr phase in their life is when they are their most territorial the behaviour in combination with persistence indicate a higher level of aggression. (Näslund and Johnsson, 2016; Elwood et al., 2014) 

Figure 1: Remove 0 in the 0-5min start box otherwise need to say 5-10 etc. for the other timelines. Also, what the numbers in brackets refers to? Shouldn’t it be “-5 to 0” for the open field and “0 to 20” for the mirror?

Reply: Yes, thank you for you feedback, this has now been changed in the figure. 

Figure 2: Can you add a line with arrows that indicates the start and end of the arena measurements? As it is now it’s not clear to which exact part of the arena 42cm refers to.

Reply: Yes this has been corrected, thank you. 

Figure 4: Is this the distance moved per minute? Please make it clearer in the figure as well as in methods and results.

Reply: Thank you for your comment. The line “The X-axis represents the accumulated amount for each minute. ” has been added to the figure legend in order to clarify the used unit. 

Figure 5: Panel labels are missing which make very difficult to understand which part of the figure results refers to.

Reply: Thank you, we have clarified the panel labels in the caption of Figure 5 at line 377-378. 

Reviewer #3

This manuscript seeks to develop a high-throughput method to screen aggressive behavioural traits in Baltic salmon because of the deleterious effects these fish can have in Aquaculture settings. Characterising aggressive traits can be time-consuming and labour intensive and so the authors investigate whether other easily measured traits, such as boldness and activity, can be used as a proxy for aggression. The authors use open field and mirror imaging tests and assign nearly 2000 fish individual activity, boldness and lateralization scores. On a subset of ~100 fish, they manually assign aggression scores (based on mirror strikes) and from this, they successfully extrapolate aggression scores to the remaining fish. They find no clear relationship between boldness/activity and aggression and head lateralization and aggression but do find a relationship between boldness and activity.

I find the overarching aims and findings of the study to be really interesting and the methods and sample sizes to be robust. I therefore firmly believe the study will make a great contribution to the literature. However, in its current form, I feel as if readers will be lost during the results and discussion sections and the findings and story of the manuscript will be overlooked. I understand the difficulties in displaying lengthy results but I encourage the authors to significantly reduce the text of the results and make their key findings a little clearer. For example, perhaps a portion of this could be done by using tables to display estimated marginal mean comparisons. The discussion also needs slight editing and trimming (i.e. lengthy paragraphs) as I feel it loses focus of the study aims and the key findings. Apart from that, I commend the authors on a great study and provide a few more comments/concerns below. I hope the authors find my suggestions constructive in improving their manuscript.

Reply: The authors thank the reviewer 3 for the insightful comment and great feedback. 

Introduction

A well written and clear introduction – really nice!

Reply: Thank you

Methods

Lines 126-137: Include how many arenas you had somewhere here.

Reply: Thank you this as been corrected. 

Line 130: Remove “from the river Daläven” as you mention this in line 136.

Reply: Thank you this has been removed. 

Line 137: Include that you recorded temperature for each trial so the reader is less concerned about the large difference in temps.

Reply: Thank you this has been added to the line 151-154. 

Line 144: Does this mean there were 4 fish in each arena? Just needs to be made a little clearer.

Reply: Thank you this has been changed on line 161. 

Line 157: Perhaps provide how many trials were performed as well.

Reply: Thank you for your feedback, we chose to report rather how many fish that were used as sample size rather then number of trials since the number of trials is not as indicative of the number of fish that contributed to the sample size (line 157-158).

Line 185: Can remove the definition of striking here as you mention it not far above.

Reply: Yes, thank you this has been removed. 

Line 187: maybe include “Striking for 50-60 sec/min” here.

Reply: Thank you for your suggestions this has been added on line 202.

Fig 3: Provide y-axis lines and in the caption for A) mention that the colour scale was for the subset of fish manually scored and the grey dots represents all fish from the study.

Reply: Yes, thank you. This line “The colored dots represent the subset of fish manually scored and the grey dots represents all fish from the study” has been added in the figure legend; 

Line 207: It would be beneficial to include these model results in a supplementary file.

Reply: We feel this will add too many models to the supplementary information, which will not improve readability. Instead, we will upload the raw data to fig share and leave it up to the reader to investigate the effect of for example session.

Line 219: Any reason why a random effect of temperature was not used here?

Reply: While temperature directly affected distance moved in arena, duration moving in arena and distance moved in mirror zone, there was no clear effect on temperature on duration in centre zone (boldness). (line 241-243)

Line 223: How did you assess the significance of factors? Through the model summary or did you run an anova on the model? i.e. code in R would be anova(mod1) to assess the significance of main factors.

Reply: We used the R command anova throughout. 

Line 226: This is a little unclear. Does this mean correlations were just for the data from the 1st minute and 5th minute?

Reply: Yes, that is correct. This is also indicated in Table 2 where the correlation coefficients are presented.

Results

I encourage the authors to make their data publically available on a platform like figshare.

Reply: Thank you for this suggestion. Yes, we will make the data publicly available on figshare once the manuscript is accepted for publication. 

Line 245 and throughout: When mentioning each variable like “The activity variable duration moving in arena…” perhaps put ‘duration moving in arena’ in quotation marks to help with the ease of reading.

Reply: Thank you for this suggestion. We have added quotation marks.

Line 251: The panel “G” is missing in Fig 4.

Reply: Thank you, this was a remnant from an older version. This line is removed from the figure caption. 

Line 270/Fig 4: I suggest you provide small subheadings above each panel/ pair of panels to indicate what these represent (i.e. “Activity” or “Boldness”). It might also be worthwhile including boxplots of overall average scores for each aggression level next to these plots.

Reply: We have taken this suggestion into consideration but we have decided to keep the figures simple and not add subheadings. However, we have adapted the Figure 4 caption to include activity, boldness and aggression behind each variable. 

The time component is so crucial to interpret the behavior, because of the removal of the hatches and hence the start of a different test plus interference by the experimenter. Therefore we do not provide average scores over time per aggression group.

Lines 331-332: Maybe remind the reader that ‘frequency of entries into the center zone’ refers to boldness.

Reply: Yes, thank you this has been changed on line 350. 

Lines 348-376: I think you mean “GLMM” here and not “LMM”.

Reply: In the paragraphs about head direction, it should say “GLM contrasts” rather than “LMM contrasts”. We changed this in line 366-396. Thank you for spotting this!

Lines 348-376: Can the results of the LMM contrasts not go into a table?

Reply: It is possible to do so, however, we argue that the information in the table is enough for giving information about the results and giving base for our conclusions. Adding more information will make the table harder to grasp.

Discussion

Line 378-397: I suggest you first refer back to the study aims and then continue with key findings. Although there was no relationship found between aggression and boldness/activity, the study methods are robust and the findings are useful so this needs to be driven home in the opening paragraph here.

Reply: We have added an introductory paragraph to the discussion (line 398-409)

Line 414: Maybe break this up into two paragraphs.

Reply: Thank you this paragraph has been separated into two sections. 

Line 455-456: Your finding of a strong relationship between boldness and activity is really interesting, especially with respect to wild passive fisheries targeting bold and active phenotypes. Perhaps a paragraph here on this would broaden the audience of your manuscript.

Reply: Thank you. We agree that relationship between activity and boldness is interesting. It has been shown repeatedly that bold, active phenotypes are more likely to be caught by passive fisheries than shy fish (e.g. Koeck et al., 2019; DOI: 10.1139/cjfas-2018-0085). Sea ranching appears to result in an unintentional selection for bold active fish, an effect which may be further complicated by passive fisheries (hook and line fishing at sea and sportfishing on salmon returning to the river). However, there are no fishing on salmon at the smolt stage but smolts leaving the river entering the estuary suffer from heavy piscean and avian predation. Again, bold individuals may suffer from higher predation than shy ones. Thus, the sea ranching related selection for bold phenotypes may be counteracted by predation and fishery. This is, however, a highly speculative suggestion. Even though interesting, we feel that a lengthily discussion on the effects of fishery and predation is out of the scope of our study. 

Line 464: Remove ‘and’ from …”and to,…”

Reply: Thank you, this has been corrected. 

Line 479: Perhaps provide a few guidelines for future research.

Reply: Thank you for your comment. The following has been added in the last pharagraph: “In future studies it would be interesting to include genomics, such as genome-wide association studies (GWAS), which may make it possible to identify genetic markers for aggression.” (line 531-533)

Overall, the discussion is lacking a strong link with the aims of the study and may therefore lose the readers interest and understanding.

Reply: We hope that the changes made to the discussion has made this link clearer.

---

## [Decision Letter · Decision Letter 1]

22 Mar 2023

PONE-D-22-16761R1Boldness, activity, and aggression: insights from a large-scale study in Baltic salmon (Salmo salar L)PLOS ONE

Dear Prof. Winberg,

Thank you for submitting your manuscript to PLOS ONE. After careful consideration, we feel that it has merit but does not fully meet PLOS ONE’s publication criteria as it currently stands. Therefore, we invite you to submit a revised version of the manuscript that addresses the points raised during the review process.

If on the one hand all the reviewers acknowledged that the manuscript has been significantly improved, on the other they underline that the temperature factor was not considered in the behavior model. So I invite the authors to discuss this point in more detail.

We look forward to receiving your revised manuscript.

Kind regards,

Pierluigi Carbonara, PhD

Academic Editor

PLOS ONE

Journal Requirements:

Reviewers' comments:

Reviewer's Responses to Questions

**Comments to the Author**

1. If the authors have adequately addressed your comments raised in a previous round of review and you feel that this manuscript is now acceptable for publication, you may indicate that here to bypass the “Comments to the Author” section, enter your conflict of interest statement in the “Confidential to Editor” section, and submit your "Accept" recommendation.

Reviewer #1: All comments have been addressed

Reviewer #2: All comments have been addressed

Reviewer #3: All comments have been addressed

2. Is the manuscript technically sound, and do the data support the conclusions?

Reviewer #1: Yes

Reviewer #2: Yes

Reviewer #3: Yes

3. Has the statistical analysis been performed appropriately and rigorously? 

Reviewer #1: Yes

Reviewer #2: Yes

Reviewer #3: Yes

4. Have the authors made all data underlying the findings in their manuscript fully available?

Reviewer #1: Yes

Reviewer #2: No

Reviewer #3: Yes

5. Is the manuscript presented in an intelligible fashion and written in standard English?

Reviewer #1: Yes

Reviewer #2: Yes

Reviewer #3: Yes

6. Review Comments to the Author

Reviewer #1: The authors answer my concerns and they carefully revised the manuscript following my comments and those from other reviewers. I feel that the manuscript has been greatly improved and is ready for publication. I only have two minor comments regarding the revised version, please see below:

L61-64: there is a mix of concept between personalities and coping styles in these lines (which is not the case in the following lines where it is clearer). So, please clarify the lines 61-64.

L503-509: I overall agree with the authors statements regarding the difficulties for measuring consistency of behavioral response in aquaculture and the causes of such difficulty (life stages, size, environmental parameters). Please add some reference to support the statements.

Reviewer #2: The authors have done a great work addressing most points raised and the manuscript. However, in the version I received the figures were in a very bad quality and it was impossible to check that the requested changes on the figures have been made. In addition, here below a few specific comments.

L219: Why temperature was removed from this model?

Reply: we decided to not include temperature after extensive visual exploration of the data. In contrast to locomotory variables distance moved in arena, proportion of time moving in arena and distance moved in mirror zone, there was no clear effect of temperature on the proportion of time spent in the center zone or mirror zone. The effect of temperature on locomotory behavior of fish is also well established in the literature [54](line 241-243).

Thank you for clarifying the reason why temperature was excluded from those models. I would only suggest to remove the sentence in line 242: “Morover the effect of temperature on locomotory behaviour of fish is also well established in literature [54]” because not relevant to the previous statement. The use of space, such as center zone and mirror zone, is not necessarily strictly related to locomotion behaviour, in addition, if there was an effect of temperature on locomotion behaviour then temperature should be included and finally, the the reference seems to be about a study that has been done on growth and not on locomotion behaviour.

Line 111: Not clear what response to boldness at group level means. In addition, I think this paragraph should be discussed and integrated in the previous paragraph when presenting and discussing the objectives of the study.

Reviewer #3: The authors have addressed my comments appropriately and the manuscript has improved. I just have two more queries.

Line 157 – 158: Provide trial numbers along with total sample size.

Line 241-243: I’m needing more assurance that temperature did not have a confounding effect on boldness. An ~11 ˚C temperature differentiation over the experiment is significant and therefore potential temperature effects need to be substantially checked and accounted for. I see you mention to another reviewer that the effects were "visually assessed" however it wouldn't be too time-consuming to compare the explanatory of models with and without temperature. I suggest this is done. I also advice you provide reviewers with the raw data so that we can help with tasks like these.

7. PLOS authors have the option to publish the peer review history of their article (what does this mean?). If published, this will include your full peer review and any attached files.

Reviewer #1: No

Reviewer #2: No

Reviewer #3: No

---

## [Author Response · Author response to Decision Letter 1]

5 Jun 2023

Reviewers' comments:

Reviewer's Responses to Questions

Comments to the Author

1. If the authors have adequately addressed your comments raised in a previous round of review and you feel that this manuscript is now acceptable for publication, you may indicate that here to bypass the “Comments to the Author” section, enter your conflict of interest statement in the “Confidential to Editor” section, and submit your "Accept" recommendation.

Reviewer #1: All comments have been addressed

Reviewer #2: All comments have been addressed

Reviewer #3: All comments have been addressed

2. Is the manuscript technically sound, and do the data support the conclusions?

Reviewer #1: Yes

Reviewer #2: Yes

Reviewer #3: Yes

3. Has the statistical analysis been performed appropriately and rigorously?

Reviewer #1: Yes

Reviewer #2: Yes

Reviewer #3: Yes

4. Have the authors made all data underlying the findings in their manuscript fully available?

Reviewer #1: Yes

Reviewer #2: No

Raw data has been included in the submission of the revised version.

Reviewer #3: Yes

5. Is the manuscript presented in an intelligible fashion and written in standard English?

Reviewer #1: Yes

Reviewer #2: Yes

Reviewer #3: Yes

6. Review Comments to the Author

Reviewer #1: The authors answer my concerns and they carefully revised the manuscript following my comments and those from other reviewers. I feel that the manuscript has been greatly improved and is ready for publication. I only have two minor comments regarding the revised version, please see below:

L61-64: there is a mix of concept between personalities and coping styles in these lines (which is not the case in the following lines where it is clearer). So, please clarify the lines 61-64.

The text has been changed to: “It has becoming increasingly clear that salmonids and other teleosts display intraspecific divergence in behavioral and physiological responses to challenge [23]. Behavioral responses is often described as personalities, which include boldness, exploration, activity, aggression, and sociability [24]. Physiological responses are often called stress coping and includes physiological traits, such as sympathetic reactivity and post-stress plasma cortisol [25].” (line 62-66)

L503-509: I overall agree with the authors statements regarding the difficulties for measuring consistency of behavioural response in aquaculture and the causes of such difficulty (life stages, size, environmental parameters). Please add some reference to support the statements.

Thank you for your comment. In order to support our statements, the following references was added: Stamps and Grothius (2010)

Reviewer #2: The authors have done a great work addressing most points raised and the manuscript. However, in the version I received the figures were in a very bad quality and it was impossible to check that the requested changes on the figures have been made. In addition, here below a few specific comments.

We are sorry about the bad quality of the figures you received. However, the figures submitted submission portal were in high resolution and when checking submitted files we found that they were of good quality and clearly readable. However, we fully agree that in the manuscript the quality is very bad. Thus, something is happening during the submission process which is not in our hands to control.

L219: Why temperature was removed from this model?

Reply: we decided to not include temperature after extensive visual exploration of the data. In contrast to locomotory variables distance moved in arena, proportion of time moving in arena and distance moved in mirror zone, there was no clear effect of temperature on the proportion of time spent in the center zone or mirror zone. The effect of temperature on locomotory behavior of fish is also well established in the literature [54](line 241-243).

Thank you for clarifying the reason why temperature was excluded from those models. I would only suggest to remove the sentence in line 242: “Morover the effect of temperature on locomotory behaviour of fish is also well established in literature [54]” because not relevant to the previous statement. The use of space, such as center zone and mirror zone, is not necessarily strictly related to locomotion behaviour, in addition, if there was an effect of temperature on locomotion behaviour then temperature should be included and finally, the the reference seems to be about a study that has been done on growth and not on locomotion behaviour.

“Thank you for your comment, the line 242 has now been removed”

Line 111: Not clear what response to boldness at group level means. In addition, I think this paragraph should be discussed and integrated in the previous paragraph when presenting and discussing the objectives of the study.

“Thank you for your comment. Boldness on a group level refers to a test where the individual results is not tracked but the results is taken into an account for the whole group. In order to clarify the above statement. In an attempt to make this more clear the text ahs been changed: “Behavioral tests at group level, e.g. response to hypoxia, boldness, have been described for several teleosts. However, when testing fish in groups behavioral responses may be confound by agonistic interactions. Moreover, in salmon there is limited information on how responses to hypoxia and/or boldness determined in groups relates to aggressive behavior of individual fish.” (line 108-111).

Reviewer #3: The authors have addressed my comments appropriately and the manuscript has improved. I just have two more queries.

Line 157 – 158: Provide trial numbers along with total sample size. 

In total we performed 137 trials. This information has now has been added (line 164).

Line 241-243: I’m needing more assurance that temperature did not have a confounding effect on boldness. An ~11 ˚C temperature differentiation over the experiment is significant and therefore potential temperature effects need to be substantially checked and accounted for. I see you mention to another reviewer that the effects were "visually assessed" however it wouldn't be too time-consuming to compare the explanatory of models with and without temperature. I suggest this is done. I also advice you provide reviewers with the raw data so that we can help with tasks like these.

We found no effect of temperature on time spent in the center zone or mirror zone and the text has been changed to:“…although here we did not include a random effect of temperature since there was no clear effect of temperature on the proportion of time spent in the center zone (r2=0.0035) or mirror zone (r2=0.0017). In both cases we used pooled within-test-minute correlations. Despite those values were significant (p<0.001), we regarded the level of significance as inflated due to multiple comparison, and in addition, the effect size were regarded as to low to be relevant.” (line 240-245). 

7. PLOS authors have the option to publish the peer review history of their article (what does this mean?). If published, this will include your full peer review and any attached files.

Do you want your identity to be public for this peer review? For information about this choice, including consent withdrawal, please see our Privacy Policy.

Reviewer #1: No

Reviewer #2: No

Reviewer #3: No

 Reviewers' comments:

Reviewer's Responses to Questions

Comments to the Author

1. If the authors have adequately addressed your comments raised in a previous round of review and you feel that this manuscript is now acceptable for publication, you may indicate that here to bypass the “Comments to the Author” section, enter your conflict of interest statement in the “Confidential to Editor” section, and submit your "Accept" recommendation.

Reviewer #1: All comments have been addressed

Reviewer #2: All comments have been addressed

Reviewer #3: All comments have been addressed

2. Is the manuscript technically sound, and do the data support the conclusions?

Reviewer #1: Yes

Reviewer #2: Yes

Reviewer #3: Yes

3. Has the statistical analysis been performed appropriately and rigorously?

Reviewer #1: Yes

Reviewer #2: Yes

Reviewer #3: Yes

4. Have the authors made all data underlying the findings in their manuscript fully available?

Reviewer #1: Yes

Reviewer #2: No

Raw data has been included in the submission of the revised version.

Reviewer #3: Yes

5. Is the manuscript presented in an intelligible fashion and written in standard English?

Reviewer #1: Yes

Reviewer #2: Yes

Reviewer #3: Yes

6. Review Comments to the Author

Reviewer #1: The authors answer my concerns and they carefully revised the manuscript following my comments and those from other reviewers. I feel that the manuscript has been greatly improved and is ready for publication. I only have two minor comments regarding the revised version, please see below:

L61-64: there is a mix of concept between personalities and coping styles in these lines (which is not the case in the following lines where it is clearer). So, please clarify the lines 61-64.

The text has been changed to: “It has becoming increasingly clear that salmonids and other teleosts display intraspecific divergence in behavioral and physiological responses to challenge [23]. Behavioral responses is often described as personalities, which include boldness, exploration, activity, aggression, and sociability [24]. Physiological responses are often called stress coping and includes physiological traits, such as sympathetic reactivity and post-stress plasma cortisol [25].” (line 62-66)

L503-509: I overall agree with the authors statements regarding the difficulties for measuring consistency of behavioural response in aquaculture and the causes of such difficulty (life stages, size, environmental parameters). Please add some reference to support the statements.

Thank you for your comment. In order to support our statements, the following references was added: Stamps and Grothius (2010)

Reviewer #2: The authors have done a great work addressing most points raised and the manuscript. However, in the version I received the figures were in a very bad quality and it was impossible to check that the requested changes on the figures have been made. In addition, here below a few specific comments.

We are sorry about the bad quality of the figures you received. However, the figures submitted submission portal were in high resolution and when checking submitted files we found that they were of good quality and clearly readable. However, we fully agree that in the manuscript the quality is very bad. Thus, something is happening during the submission process which is not in our hands to control.

L219: Why temperature was removed from this model?

Reply: we decided to not include temperature after extensive visual exploration of the data. In contrast to locomotory variables distance moved in arena, proportion of time moving in arena and distance moved in mirror zone, there was no clear effect of temperature on the proportion of time spent in the center zone or mirror zone. The effect of temperature on locomotory behavior of fish is also well established in the literature [54](line 241-243).

Thank you for clarifying the reason why temperature was excluded from those models. I would only suggest to remove the sentence in line 242: “Morover the effect of temperature on locomotory behaviour of fish is also well established in literature [54]” because not relevant to the previous statement. The use of space, such as center zone and mirror zone, is not necessarily strictly related to locomotion behaviour, in addition, if there was an effect of temperature on locomotion behaviour then temperature should be included and finally, the the reference seems to be about a study that has been done on growth and not on locomotion behaviour.

“Thank you for your comment, the line 242 has now been removed”

Line 111: Not clear what response to boldness at group level means. In addition, I think this paragraph should be discussed and integrated in the previous paragraph when presenting and discussing the objectives of the study.

“Thank you for your comment. Boldness on a group level refers to a test where the individual results is not tracked but the results is taken into an account for the whole group. In order to clarify the above statement. In an attempt to make this more clear the text ahs been changed: “Behavioral tests at group level, e.g. response to hypoxia, boldness, have been described for several teleosts. However, when testing fish in groups behavioral responses may be confound by agonistic interactions. Moreover, in salmon there is limited information on how responses to hypoxia and/or boldness determined in groups relates to aggressive behavior of individual fish.” (line 108-111).

Reviewer #3: The authors have addressed my comments appropriately and the manuscript has improved. I just have two more queries.

Line 157 – 158: Provide trial numbers along with total sample size. 

In total we performed 137 trials. This information has now has been added (line 164).

Line 241-243: I’m needing more assurance that temperature did not have a confounding effect on boldness. An ~11 ˚C temperature differentiation over the experiment is significant and therefore potential temperature effects need to be substantially checked and accounted for. I see you mention to another reviewer that the effects were "visually assessed" however it wouldn't be too time-consuming to compare the explanatory of models with and without temperature. I suggest this is done. I also advice you provide reviewers with the raw data so that we can help with tasks like these.

We found no effect of temperature on time spent in the center zone or mirror zone and the text has been changed to:“…although here we did not include a random effect of temperature since there was no clear effect of temperature on the proportion of time spent in the center zone (r2=0.0035) or mirror zone (r2=0.0017). In both cases we used pooled within-test-minute correlations. Despite those values were significant (p<0.001), we regarded the level of significance as inflated due to multiple comparison, and in addition, the effect size were regarded as to low to be relevant.” (line 240-245). 

7. PLOS authors have the option to publish the peer review history of their article (what does this mean?). If published, this will include your full peer review and any attached files.

Do you want your identity to be public for this peer review? For information about this choice, including consent withdrawal, please see our Privacy Policy.

Reviewer #1: No

Reviewer #2: No

Reviewer #3: No

 Reviewers' comments:

Reviewer's Responses to Questions

Comments to the Author

1. If the authors have adequately addressed your comments raised in a previous round of review and you feel that this manuscript is now acceptable for publication, you may indicate that here to bypass the “Comments to the Author” section, enter your conflict of interest statement in the “Confidential to Editor” section, and submit your "Accept" recommendation.

Reviewer #1: All comments have been addressed

Reviewer #2: All comments have been addressed

Reviewer #3: All comments have been addressed

2. Is the manuscript technically sound, and do the data support the conclusions?

Reviewer #1: Yes

Reviewer #2: Yes

Reviewer #3: Yes

3. Has the statistical analysis been performed appropriately and rigorously?

Reviewer #1: Yes

Reviewer #2: Yes

Reviewer #3: Yes

4. Have the authors made all data underlying the findings in their manuscript fully available?

Reviewer #1: Yes

Reviewer #2: No

Raw data has been included in the submission of the revised version.

Reviewer #3: Yes

5. Is the manuscript presented in an intelligible fashion and written in standard English?

Reviewer #1: Yes

Reviewer #2: Yes

Reviewer #3: Yes

6. Review Comments to the Author

Reviewer #1: The authors answer my concerns and they carefully revised the manuscript following my comments and those from other reviewers. I feel that the manuscript has been greatly improved and is ready for publication. I only have two minor comments regarding the revised version, please see below:

L61-64: there is a mix of concept between personalities and coping styles in these lines (which is not the case in the following lines where it is clearer). So, please clarify the lines 61-64.

The text has been changed to: “It has becoming increasingly clear that salmonids and other teleosts display intraspecific divergence in behavioral and physiological responses to challenge [23]. Behavioral responses is often described as personalities, which include boldness, exploration, activity, aggression, and sociability [24]. Physiological responses are often called stress coping and includes physiological traits, such as sympathetic reactivity and post-stress plasma cortisol [25].” (line 62-66)

L503-509: I overall agree with the authors statements regarding the difficulties for measuring consistency of behavioural response in aquaculture and the causes of such difficulty (life stages, size, environmental parameters). Please add some reference to support the statements.

Thank you for your comment. In order to support our statements, the following references was added: Stamps and Grothius (2010)

Reviewer #2: The authors have done a great work addressing most points raised and the manuscript. However, in the version I received the figures were in a very bad quality and it was impossible to check that the requested changes on the figures have been made. In addition, here below a few specific comments.

We are sorry about the bad quality of the figures you received. However, the figures submitted submission portal were in high resolution and when checking submitted files we found that they were of good quality and clearly readable. However, we fully agree that in the manuscript the quality is very bad. Thus, something is happening during the submission process which is not in our hands to control.

L219: Why temperature was removed from this model?

Reply: we decided to not include temperature after extensive visual exploration of the data. In contrast to locomotory variables distance moved in arena, proportion of time moving in arena and distance moved in mirror zone, there was no clear effect of temperature on the proportion of time spent in the center zone or mirror zone. The effect of temperature on locomotory behavior of fish is also well established in the literature [54](line 241-243).

Thank you for clarifying the reason why temperature was excluded from those models. I would only suggest to remove the sentence in line 242: “Morover the effect of temperature on locomotory behaviour of fish is also well established in literature [54]” because not relevant to the previous statement. The use of space, such as center zone and mirror zone, is not necessarily strictly related to locomotion behaviour, in addition, if there was an effect of temperature on locomotion behaviour then temperature should be included and finally, the the reference seems to be about a study that has been done on growth and not on locomotion behaviour.

“Thank you for your comment, the line 242 has now been removed”

Line 111: Not clear what response to boldness at group level means. In addition, I think this paragraph should be discussed and integrated in the previous paragraph when presenting and discussing the objectives of the study.

“Thank you for your comment. Boldness on a group level refers to a test where the individual results is not tracked but the results is taken into an account for the whole group. In order to clarify the above statement. In an attempt to make this more clear the text ahs been changed: “Behavioral tests at group level, e.g. response to hypoxia, boldness, have been described for several teleosts. However, when testing fish in groups behavioral responses may be confound by agonistic interactions. Moreover, in salmon there is limited information on how responses to hypoxia and/or boldness determined in groups relates to aggressive behavior of individual fish.” (line 108-111).

Reviewer #3: The authors have addressed my comments appropriately and the manuscript has improved. I just have two more queries.

Line 157 – 158: Provide trial numbers along with total sample size. 

In total we performed 137 trials. This information has now has been added (line 164).

Line 241-243: I’m needing more assurance that temperature did not have a confounding effect on boldness. An ~11 ˚C temperature differentiation over the experiment is significant and therefore potential temperature effects need to be substantially checked and accounted for. I see you mention to another reviewer that the effects were "visually assessed" however it wouldn't be too time-consuming to compare the explanatory of models with and without temperature. I suggest this is done. I also advice you provide reviewers with the raw data so that we can help with tasks like these.

We found no effect of temperature on time spent in the center zone or mirror zone and the text has been changed to:“…although here we did not include a random effect of temperature since there was no clear effect of temperature on the proportion of time spent in the center zone (r2=0.0035) or mirror zone (r2=0.0017). In both cases we used pooled within-test-minute correlations. Despite those values were significant (p<0.001), we regarded the level of significance as inflated due to multiple comparison, and in addition, the effect size were regarded as to low to be relevant.” (line 240-245). 

7. PLOS authors have the option to publish the peer review history of their article (what does this mean?). If published, this will include your full peer review and any attached files.

Do you want your identity to be public for this peer review? For information about this choice, including consent withdrawal, please see our Privacy Policy.

Reviewer #1: No

Reviewer #2: No

Reviewer #3: No

---

## [Editor Report · Decision Letter 2]

14 Jun 2023

Boldness, activity, and aggression: insights from a large-scale study in Baltic salmon (Salmo salar L)

PONE-D-22-16761R2

Dear Dr. Winberg,

We’re pleased to inform you that your manuscript has been judged scientifically suitable for publication and will be formally accepted for publication once it meets all outstanding technical requirements.

Kind regards,

Pierluigi Carbonara, PhD

Academic Editor

PLOS ONE
---

## [Editor Report · Acceptance letter]

10 Jul 2023

PONE-D-22-16761R2 

Boldness, activity, and aggression: insights from a large-scale study in Baltic salmon (Salmo salar L) 

Dear Dr. Winberg:

I'm pleased to inform you that your manuscript has been deemed suitable for publication in PLOS ONE. Congratulations! Your manuscript is now with our production department. 

Kind regards, 

on behalf of

Dr. Pierluigi Carbonara 

Academic Editor

PLOS ONE